# Dehydration of subducting slow-spread oceanic lithosphere in the Lesser Antilles

Michele Paulatto[1,2], Mireille Laigle[1], Audrey Galve[1], Philippe Charvis[1], Martine Sapin[3], Gaye Bayrakci[4], Mikael Evain[5] & Heidrun Kopp[6]

Subducting slabs carry water into the mantle and are a major gateway in the global geochemical water cycle. Fluid transport and release can be constrained with seismological data. Here we use joint active-source/local-earthquake seismic tomography to derive unprecedented constraints on multi-stage fluid release from subducting slow-spread oceanic lithosphere. We image the low P-wave velocity crustal layer on the slab top and show that it disappears beneath 60–100 km depth, marking the depth of dehydration metamorphism and eclogitization. Clustering of seismicity at 120–160 km depth suggests that the slab's mantle dehydrates beneath the volcanic arc, and may be the main source of fluids triggering arc magma generation. Lateral variations in seismic properties on the slab surface suggest that serpentinized peridotite exhumed in tectonized slow-spread crust near fracture zones may increase water transport to sub-arc depths. This results in heterogeneous water release and directly impacts earthquakes generation and mantle wedge dynamics.

[1] Université Côte d'Azur, CNRS, IRD, Observatoire de la Côte d'Azur, Géoazur, 250 rue Albert Einstein, Sophia Antipolis, 06560 Valbonne, France. [2] Imperial College London, Earth Science and Engineering, Prince Consort Road, London SW7 2BP, UK. [3] Institut de Physique du Globe de Paris, PRES Sorbonne Paris-Cité, CNRS UMR 7154, 1 rue Jussieu, 75005 Paris, France. [4] University of Southampton, Ocean and Earth Science, European Way, Southampton SO14 3ZH, UK. [5] IFREMER, Géosciences Marines, Centre Bretagne, ZI de la Pointe du Diable–CS 10070, 29280 Plouzané, France. [6] GEOMAR Helmholtz Centre for Ocean Research Kiel, Wischhofstr. 1-3, 24148 Kiel, Germany. Correspondence and requests for materials should be addressed to M.P. (email: m.paulatto@imperial.ac.uk) or to M.L. (email: laigle@geoazur.unice.fr).

The oceanic plate is extensively hydrated by interaction with seawater throughout its life from the mid-ocean ridge to the trench[1]. The mafic crust can store up to 5 wt% structurally bound water (water present in the structure of minerals) and an additional ~5 wt% pore water[2]. A significant amount of water can also penetrate into the upper mantle[1], particularly through outer rise bend faults, where it bounds with peridotites to form serpentine. At subduction zones, much of the stored water is released via pore fluid escape and through a series of metamorphic reactions that depend on the composition and thermal regime of the slab[1,3,4]. The most notable are eclogitization of hydrated basalt and gabbro and breakdown of serpentinite (deserpentinization). These transformations and the resulting changes in physical properties have been linked to intermediate depth seismicity, tremor and slow earthquakes[5–7]. Most constraints obtained to date are restricted to Pacific subduction zones, and have contributed to a model of slab dehydration applicable to fast-spread oceanic lithosphere with a mafic crust[3,5]. Slow-spread crust, however, is heterogeneous in thickness (ranging from 4 to 8 km) and composition[8,9] and is likely to have a different water distribution than fast-spread crust. Most importantly, slow spreading results in the exhumation and hydration of mantle peridotite at oceanic core complexes and the formation of tectonized crust, consisting of a mix of intrusive gabbros and variably serpentinized peridotite, with only a thin and patchy cover of extrusive volcanics[8,10]. Tectonized crust is particularly common near fracture zones and can result in an overall wetter lithosphere, because of the greater capacity of serpentinized peridotite to store water compared to hydrated mafic rocks[3], however, few constraints exist on the subduction of tectonized crust and its effects on seismicity, plate coupling and magmatism.

Here we study the Lesser Antilles subduction zone (LASZ), which represents the global end-member of subduction zones in terms of composition and water transport. The main characteristics are a subducting plate that was formed by slow accretion (spreading rate = 20 mm per year[11]), a cold slab due to the relatively old plate age (80–100 Myr), and a slow convergence rate (19 mm per year[12]). These characteristics and the presence of several fracture zones on the slab (Fig. 1), likely enhance water transport into the mantle[13]. Sediment thickness on the incoming plate increases to the south (from 0.5 to >4.0 km) due to the proximity of the Orinoco delta and is partitioned by a series of oblique topographic ridges (the Tiburon, Barracuda and St. Lucia ridges, Fig. 1), which sweep the accretionary wedge, likely affecting the segmentation of the megathrust fault[14]. No large ($M > 7$) thrust earthquakes have been recorded in the instrumental catalogues and only two possible large thrust events have been reported in historical records (the $M \sim 8.0$ 1839 earthquake and the $M \sim 8.5$ 1843 earthquake[15]). Modelling of GPS ground displacement[16] suggests that the margin is weakly coupled and predicts a return period for M8 events of ~2000 year. However, uncertainties are large due to the restricted distribution of GPS stations and uncertainties in slab geometry. This conundrum stands in the way of our understanding of the seismogenic behaviour of the LASZ and places considerable uncertainty on estimates of seismic and tsunami hazard.

We use joint active source/local earthquake tomography to image the subducting slab and the mantle wedge in the Lesser Antilles. Our model provides unprecedented constraints on fluid transport and release in subducting slow-spread crust and their relationship with the properties of the incoming plate and the seismogenic behaviour of the margin.

## Results

**Data selection and preprocessing.** We have carried out a three-dimensional (3D) joint seismic tomography study to image the physical properties of the crust, mantle wedge and subducting slab over a 350-km-long segment of the LASZ between Martinique and Montserrat (Fig. 1). This section was affected by the two largest thrust earthquakes[15] and is characterized by the most vigorous volcanic activity along this arc[17]. We have assembled all available active source seismic data and local earthquake data collected in this region over the last two decades into one of the largest combined traveltime databases in any subduction setting.

The active source seismic dataset includes data from several marine and amphibious experiments (EW9803 (ref. 18), Sismantilles 1 (ref. 19), TRAIL[20], Sismantilles 2 (ref. 21), SEA-CALIPSO[22]) carried out between 1998 and 2007. The complete active source traveltime dataset includes 110,807 airgun shots resulting in 461,648 P-wave traveltimes recorded on 348 seismic stations at offsets of up to 200 km. The network of stations includes ocean bottom seismometers in the forearc and land stations on the islands from Martinique to Montserrat. We also picked traveltimes on some of the permanent seismometer stations from the Montserrat Volcano Observatory (MVO) network and from the Geoscope station FDF on Martinique. Shots were decimated at a minimum inline shot spacing of 1,000 m to reduce computing load and redundancy, leading to a final selection of 8,097 shots and 71,568 P-wave traveltimes (Fig. 2a). Picking uncertainties were estimated qualitatively, based on signal to noise ratio, source-receiver offset, and spatial continuity of arrivals. Since the damped least squares approach used in the inversion does not allow for accurate modelling of the water layer and seabed topography we shifted the shot locations to the seabed and corrected the traveltimes assuming vertical propagation at seawater velocities.

The local earthquake data were compiled from previously published studies[23,24], using data from the Sismantilles 1, Sismantilles 2, OBSAntilles[25] and OBSISMER[26] projects, and supplemented by traveltimes recorded by the MVO network. P and S phase picks were quality checked and the events were initially relocated in a 1D model using the linearized least squares 1D joint inversion code *Velest*[27]. The resulting joint pick database contains 2,440 events in total, including 38,382 P phases and 27,092 S phases. Earthquakes with an insufficient number of picked phases (less than eight in total), large azimuthal gap ($>200°$), large RMS residual ($>1$ s) were not used in the inversion. In addition, we discarded S picks with inconsistent $T_S/T_P$ ratio (estimate of $V_P/V_S$) and stations with less than 5 observations of either phase. The resulting event catalogue was further pruned by declustering with a 2-km minimum distance between events. The final selection includes 743 events and a total of 16,277 P phases and 11,988 S phases (Fig. 2b), corresponding to an average of 22 P picks and 16 S picks per event. A larger selection of 1,660 events with less stringent selection criteria (no declustering, azimuthal gap $<270°$, RMS residual $<2$ s) was relocated in the final model after the inversion.

**Joint tomographic inversion of shots and local earthquakes.** We adopted a widely used three-dimensional joint tomography algorithm[28], which uses a pseudo-bending method to solve the forward problem[29] and a combination of parameter separation and damped least squares to tackle the inverse problem. The initial $V_P$ model (Fig. 3a) was built as a hanging 1D model (see methods) with a variable upper plate Moho, based on previous local and regional studies[20,22,30] and on 1D inversion of the earthquake data. No a priori slab anomaly was introduced. The initial $V_P/V_S$ was chosen to be 1.76 everywhere, based on the analysis of the ratio of $T_S$ to $T_P$ (the Wadati diagram, Supplementary Fig. 1).

The inversion simultaneously adjusts earthquake locations, origin times, $V_P$, $V_P/V_S$, and station corrections to reduce the data

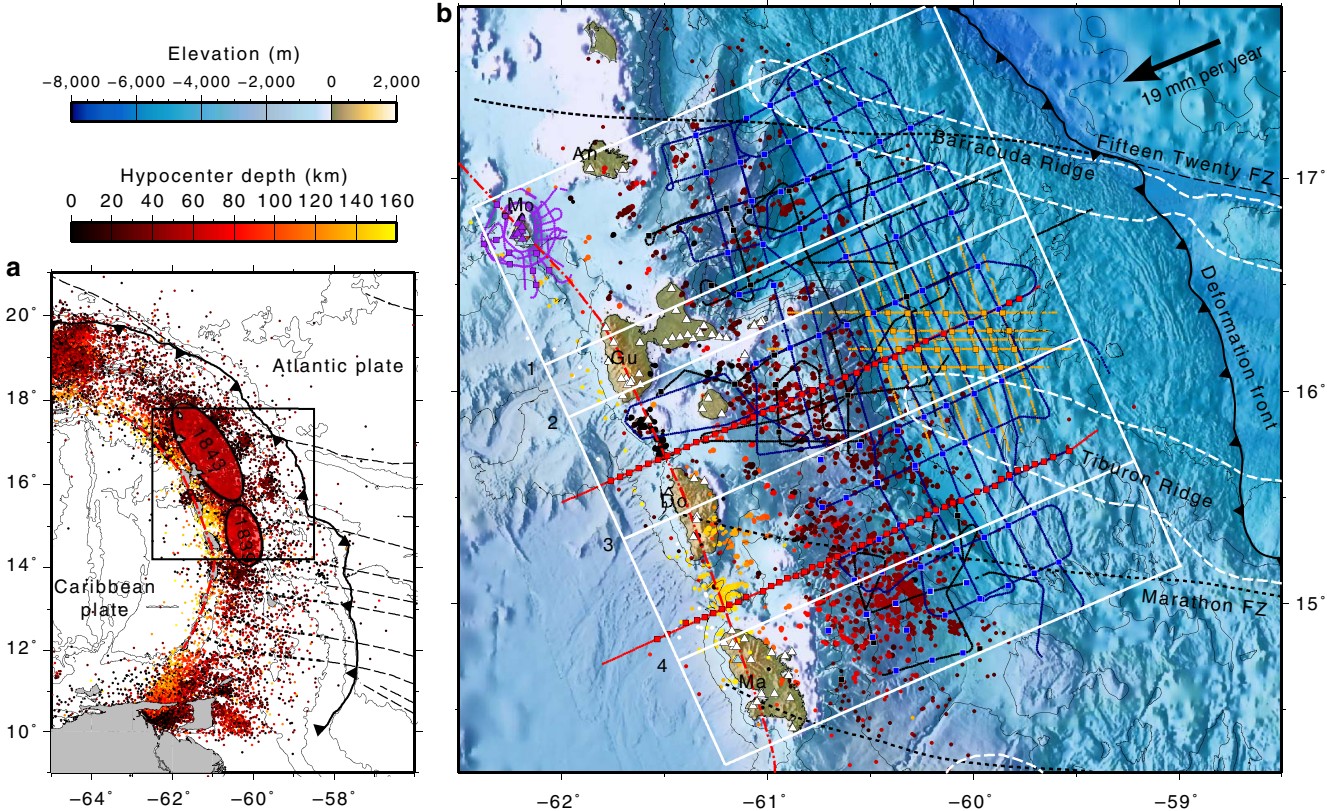

**Figure 1 | Study region and data distribution.** (**a**) Location of study area in the Lesser Antilles subduction zone. The seismicity from 1900 to 2015 from the ISC catalogue is marked by coloured dots. Ellipses mark the estimated rupture areas of the 1843 and 1839 earthquakes[15]. (**b**) Survey geometry with bathymetry of the study area. Small circles, colour-coded for depth, mark earthquake hypocenters used in this study. Smaller circles mark events that were relocated but were not used in the tomography. Coloured lines mark airgun shooting profiles from different surveys: EW9803 (orange), Sismantilles 1 (black), TRAIL (red), Sismantilles 2 (blue) and SEA-CALIPSO (purple). Triangles mark land stations. Squares mark OBSs. White dashed lines mark the outlines of the Barracuda and Tiburon ridges. Black dashed lines mark the projection onto the plate interface of the approximate prolongation of the main recognized fracture zones. The white box marks the extent of the tomography model. Numbered white lines mark the locations of profiles shown in Fig. 4. Topography and bathymetry data for this image were compiled from multi-beam bathymetry data[21,69], the GEBCO_2014 Grid, version 20150318 (www.gebco.net), and data from the ASTER Global Digital Elevation Model (GDEM) v2 (ref. 70).

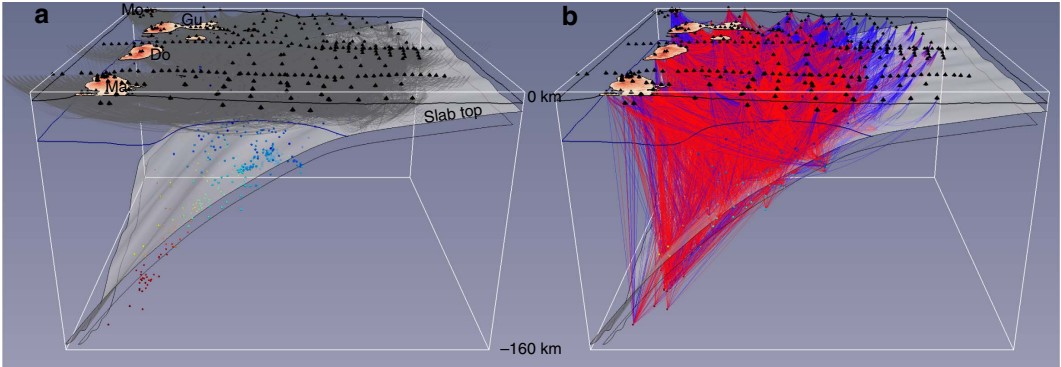

**Figure 2 | Ray coverage.** (**a**) Ray paths for active shots (P-waves). (**b**) Ray paths for local earthquakes' P-waves (blue) and S-waves (red). Black pyramids mark recording stations. Coloured dots mark earthquake hypocenters (inverted events only). The islands of Martinique (Ma), Dominica (Do), Guadeloupe (Gu) and Montserrat (Mo) as well as the location of the subducting slab are marked for reference.

misfit. The horizontal grid spacing was gradually reduced from $40 \times 50\,km$ to $15 \times 15\,km$ (Fig. 3). The vertical grid spacing increased from 3 to 5 km at the surface to 20 km at the base of the model. Multiple inversions were carried out on staggered grids to reduce the bias introduced by our choice of domain discretization. The final model was calculated as the median of the output models of the staggered inversions. The model resolution was evaluated through analysis of the resolution matrix and checkerboard tests. In the $V_P$ model the crust, mantle wedge and slab are well resolved at depth of up to 100–120 km, but the deeper parts of the slab and the mantle wedge beneath the arc likely suffer from vertical smearing. The $V_P/V_S$ model is less well resolved and suffers from stronger smearing because of fewer inverted S-phases. More details on the inversion, parameter selection and resolution assessment are described in the Methods section and Supplementary Figs 1–10.

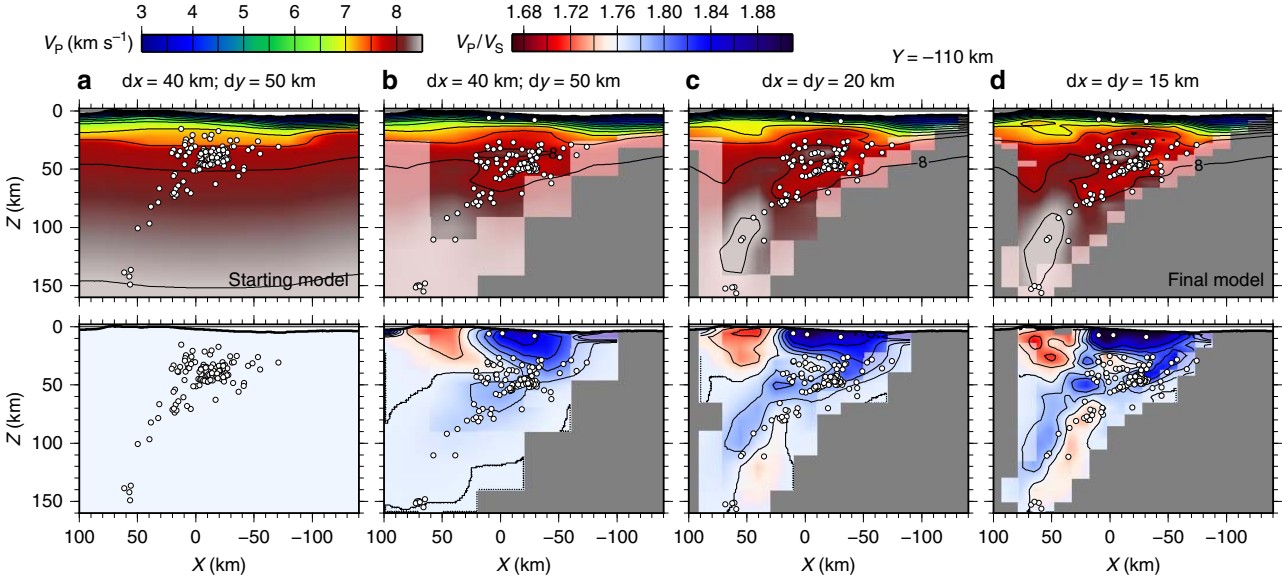

**Figure 3 | Model evolution.** Vertical cross-sections through the $V_P$ (top) and $V_P/V_S$ (bottom) models at different stages of the inversion. (**a**) Starting model. (**b**) Output of inversion with $40 \times 50$ km horizontal grid spacing. (**c**) Output of inversion with $20 \times 20$ km horizontal grid spacing. (**d**) Final model with $15 \times 15$ km horizontal grid spacing. Earthquakes are marked by white dots (inverted events only). Notice how the distribution of earthquakes becomes sharper as the inversion progresses.

The combination of shots and local earthquakes and the dense amphibious recording array allow us to sample the subduction zone in exceptional detail. The main features of the output $V_P$ and $V_P/V_S$ models are a clearly defined overriding plate crust, the mantle wedge and the subducting slab dipping westward beneath it (Figs 3 and 4).

**Geometry and structure of the subducting slab.** We observe a dipping low-$V_P$ anomaly, characterized by 5–10% lower $V_P$ than the surrounding mantle, which we interpret as the hydrated slab crust. This anomaly disappears at 60–100 km depth (Figs 3–5) and transitions into a high-$V_P$ anomaly. Slab anomaly recovery tests and slab checkerboard tests (see Methods and Supplementary Figs 7–10) show that the fading of the low-$V_P$ layer at depth is a feature required by the data, and not a consequence of diminished resolution beneath 60 km depth. Tomography images of the subducting slab crust are increasingly common[31,32] but rarely detailed in 3D. Our inversion provides one of the clearest tomographic images of the subducting slab crust and confirms evidence from other seismic observables, for example, waveguide behaviour[33–35], converted teleseismic phases[36] and receiver functions[24,37,38], that on most subduction zones a low-$V_P$ crustal layer persists to considerable depth.

We used the shape of the slab anomaly from our $V_P$ model and the seismicity distribution, in conjunction with constraints on the depth of the acoustic basement at the top of the oceanic crust from multichannel seismic reflection (MCS) profiles[39–41], to trace the approximate geometry of the slab surface from the accretionary prism to 150 km depth (Fig. 2). The slab initially dips gently at $\sim 20°$ until it reaches 50 km depth, where the dip increases to approximately 45–55°, consistently with earlier receiver function results (Fig. 4d)[24]. The overall slab geometry does not vary drastically within the survey area but the change in dip at 50 km depth appears sharper in the north than in the south.

$V_P$ on the slab surface ranges from $4.5 \text{ km s}^{-1}$ at 10 km depth to $8.5 \text{ km s}^{-1}$ at 150 km depth (Figs 5a and 6a). We observe an initial strong gradient in $V_P$ at 5 to 20 km depth and a gentler gradient beneath 20 km depth. The $V_P/V_S$ is generally higher than 1.76 in the forearc crust and within the slab, and lower in the arc

crust (Fig. 3). A similar $V_P/V_S$ distribution with high $V_P/V_S$ in the forearc crust and in the slab crust has been observed at the Hikurangi margin[42]. The $V_P/V_S$ on the slab surface decreases steadily with depth and delineates two clearly distinct domains: a shallow region characterized by moderate to high $V_P/V_S$ ($> 1.76$), and a deep region characterized by low $V_P/V_S$ ($< 1.76$) (Fig. 5c). The boundary between these two domains is sharp and is found at 50 km depth north of Dominica but seems more gradual and much deeper (close to 100 km depth) south of Dominica (Fig. 5c).

**Seismicity distribution.** Our joint inversion allows us to investigate the spatial relationship between seismicity and seismic properties in the tomographic model. Mantle wedge or supra-slab earthquakes are observed throughout the region, clustered mostly between 25 and 60 km depth (Fig. 4). They are restricted to the deep forearc crust and the shallow mantle wedge corner, extending up to 50–80 km arc-ward of the contact with the overriding plate Moho. Mantle wedge seismicity is heterogeneous and is more abundant and extends deeper in the south. It is spatially associated with moderate $V_P$ ($\sim 8.0 \text{ km s}^{-1}$) and high $V_P/V_S$ (Fig. 4). The sharp termination of seismicity towards the arc suggests it may be controlled by the thermal structure. At the Hikurangi margin mantle wedge seismicity is clustered close to the 700 °C isotherm and is thought to be induced by dehydration embrittlement caused by deserpentinization[43]. This process could explain the observed sharp termination, but in the Lesser Antilles the mantle wedge seismicity is not restricted to a narrow band and is instead observed throughout the shallow nose of the mantle wedge. This wide distribution had been noticed in earlier studies, and has been attributed to a heterogeneous mantle including volumes of pyroxenite, possibly associated with mantle plume magmatism during formation of the Caribbean plate[23,24]. Alternatively, pyroxenite may have formed by fractional crystallization of a mafic melt at the base of the crust beneath the arc[44]. Fluids from the slab are thought to be able to penetrate the low-permeability mantle wedge aided by hydrofracturing[45] and reaction-induced cracking[46]. If fluids penetrate a heterogeneous mantle wedge, veins of pyroxenite might become stressed by fluid overpressure or by volume changes caused by

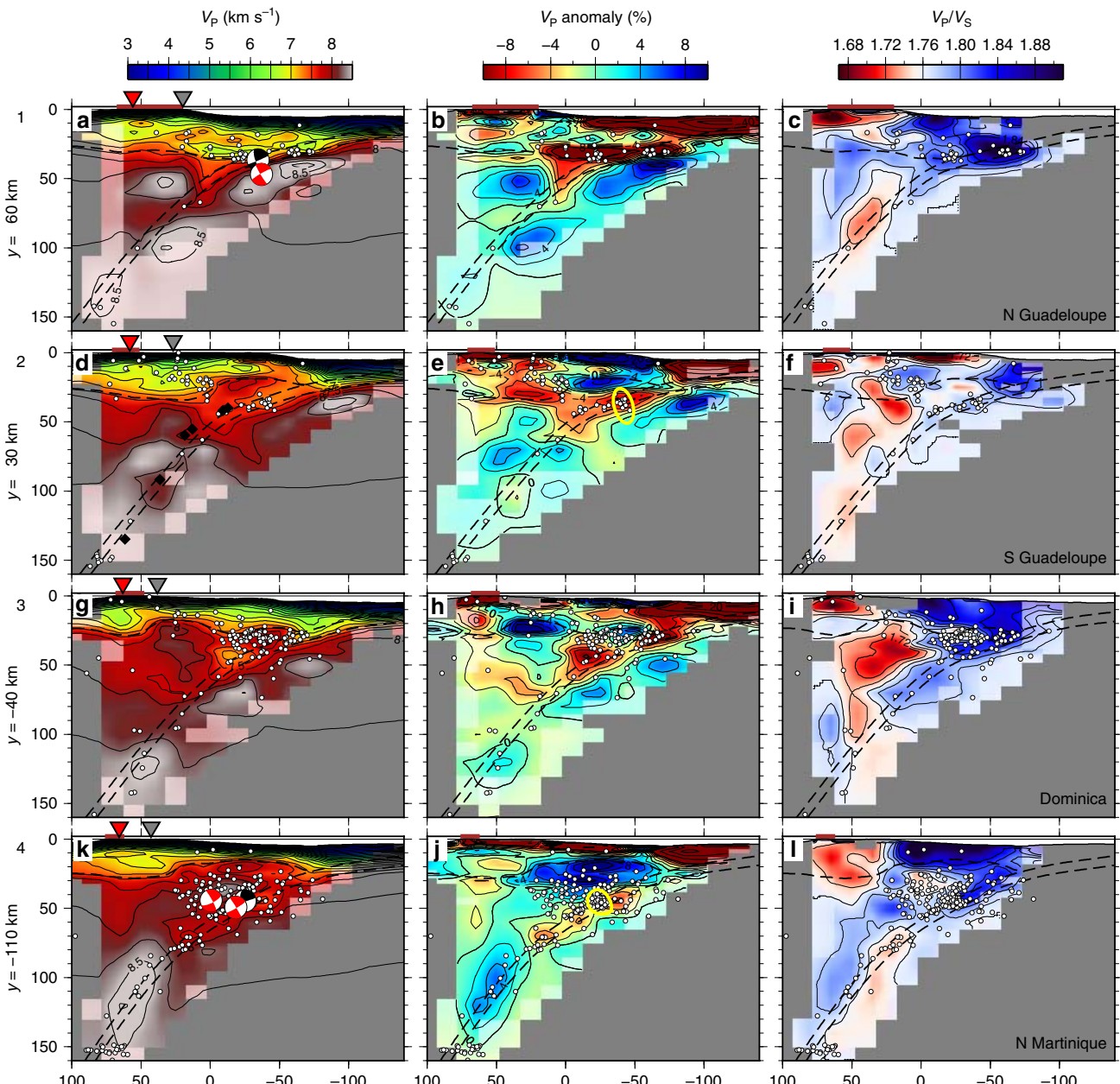

**Figure 4 | Cross sections trough the final tomography model.** (**a,d,g,k**) $V_P$. (**b,e,h,j**) $V_P$ anomaly with respect to 1D model. (**c,f,i,l**) $V_P/V_S$ ratio. Rows correspond to different profiles with locations shown in Figs 1b and 5. Dashed lines mark the interpreted location of the oceanic basement (slab top surface), the slab Moho (assuming a crustal thickness of 7 km) and the overriding plate Moho. Notice the clear slab crust low-$V_P$ anomaly and the deeper mantle wedge seismicity in the southern profile. Areas with sparse ray coverage are covered in grey. Areas with potential strong smearing are faded. Earthquake hypocenters within 15 km of each section are marked by white circles (all relocated events). Black and white focal mechanisms are shown for relocated flat thrust events. Red focal mechanisms correspond to non-relocated events from the CMT catalogue. The black diamonds in **d** mark the depth of the slab surface estimated from receiver functions[24]. The yellow ellipses in **e,j** mark dipping planar clusters of earthquakes. These are more easily identified in Supplementary Movies 1 and 2, which show a 3D rotating view of the earthquake distribution. The inverted triangles mark the location of the active volcanic arc (red) and the ancient volcanic arc (grey).

serpentinization of surrounding peridotite, and could fail seismically.

Several moderate size flat-thrust earthquakes have been observed at 40–50 km depth[24]. Our relocation places them very close to the slab surface (Fig. 4). The most recent happened on 3 February 2017 (ref. 47), with a hypocentre close to those of the $M_w$5.2 event of 6 February 2008 and the $M_w$4.9 event of 2 February 2017 (Fig. 5). These events are located within or at the edges of high $V_P$ patches, perhaps corresponding to strong asperities, supporting the hypothesis that they may be repeating

earthquakes[24], similar to those observed in northern Tohoku at the downdip limit of the seismogenic zone[48].

Intra-slab seismicity is observed mostly in the upper 10 km of the slab and delineates the Wadati-Benioff zone. A second deeper plane of seismicity located 25–30 km beneath the slab top can be distinguished only in some areas[24]. Intra-slab seismicity is densest near two clusters at 50–80 km depth and 135–155 km depth (Fig. 6c). The first cluster is likely associated with crustal dehydration reactions[5,31] and/or with stresses induced by densification and slab bending/unbending[49]. Some of this

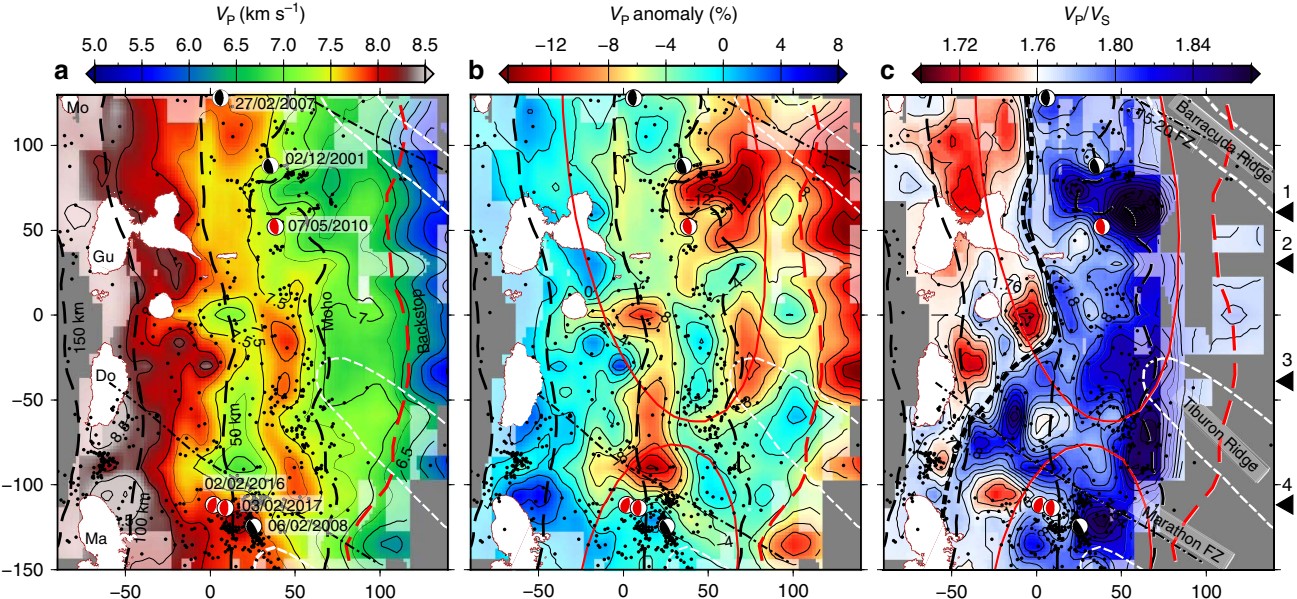

**Figure 5 | Properties of the plate interface.** Sections of the tomography model along the interpreted slab top surface showing (**a**) $V_P$, (**b**) $V_P$ anomaly and (**c**) $V_P/V_S$ ratio. Areas with sparse ray coverage are covered in grey. Areas with potential strong smearing are faded. The red dashed lines mark the interpreted location of the backstop[41]. The black dashed lines mark iso-depth contours of the slab (at 50, 100 and 150 km) and the contact of the plate interface with the overriding plate Moho. The thick black dashed line in **c** marks the transition from high to low $V_P/V_S$. The black dashed/dotted lines mark the projection of the major fracture zones on the plate interface. The white dashed lines mark the Tiburon and Barracuda ridges. The red ellipses mark the estimated rupture areas of the 1843 and 1839 earthquakes[15]. The islands (white) are overlain for reference. Earthquake hypocenters are marked by black dots (all relocated events with epicenter within the slab). Black and white focal mechanisms are shown for relocated flat thrust events. Red focal mechanisms correspond to non-relocated events from the CMT catalogue. The locations of the profiles shown in Fig. 4 are marked by black triangles on the right. Ma, Martinique; Do, Dominica; Gu, Guadeloupe; Mo, Montserrat.

seismicity appears to be distributed along steeply dipping planes penetrating ~10 km into the slab (Fig. 2b; Supplementary Movies 1 and 2) suggesting faulting of the slab or nucleation along pre-existing fault structures. The deeper cluster is located directly beneath the arc and is strongest beneath Martinique, close to the prolongation of the Marathon fracture zone (Figs 4–6). These deep events seem to originate within the slab's upper mantle and are likely related to dehydration of serpentinized peridotite.

## Discussion

The observed initial strong increase in slab surface $V_P$ at 5–20 km depth is likely to be due to dehydration of the subducted sediments and shallow crust, involving escape of pore fluids, compaction, and cementation[50,51]. Our tomography model cannot resolve the subduction channel, but MCS profiles show that it is up to 1 km thick beneath the accretionary prism[39,41]. Analysis of samples from mud volcanoes close to the deformation front indicates that fluid drainage along the decollement is limited and that diffuse flow is likely to be dominant[52]. The high $V_P/V_S$ ratio observed in the forearc crust and sedimentary basins (Fig. 4) may indicate a fractured and water-rich forearc crust, in agreement with substantial diffuse upward water flux.

The down-dip limit of the low-$V_P$ slab crust anomaly ($V_P = 8.0$ km s$^{-1}$) is found at 60–100 km depth and is relatively constant along strike (Figs 5a and 6a). This transition likely marks the metamorphism of the oceanic crust and the depth range is consistent with crustal eclogitization of a slowly subducting old cold slab[3]. A local minimum in $V_P$ at ~50 km depth, elongated in an arc-parallel direction ($V_P = $~7.5 km s$^{-1}$, Figs 5a,b and 6a), may be related to localized voluminous water release and to associated accumulation of overpressured pore fluids on the plate interface and/or serpentinization of mantle peridotites above the

slab[38,53]. This anomaly may be the seismic expression of the deep overpressure zone proposed based on seismogenic behaviour and thought to coincide with the transition between the seismogenic zone and stable shear[51]. Dehydration of the mafic crust is expected to release a significant amount of water and the strongest water release is expected from the blueschist to eclogite facies transition reactions at 300–500 °C (refs 3,54). On the basis of published thermal models of the LASZ[55,56] and assuming a mafic composition, these reactions are predicted to take place at 65–100 km depth (Fig. 6d)[57]. Dehydration of the crust at this depth range would also explain the observation of increased intra-slab seismicity at 50–80 km depth, since eclogitization has been suggested to induce stick-slip behaviour[7]. The mantle wedge above the local $V_P$ minimum is heterogeneous, and is subject to the highest rate of mantle wedge seismicity (Fig. 4), supporting the hypothesis that mantle wedge seismicity is linked to fluid fluxing from dehydration of the slab. The fact that the distribution of mantle wedge seismicity is variable along strike may indicate heterogeneous fluid release and upwelling through the cold nose of the mantle wedge, or a laterally variable thermal structure, perhaps modulated by the properties of the overriding plate. We notice in fact that the thickness of the arc and forearc crust varies significantly along strike within the survey area (Fig. 4), which may affect the dynamics of the corner flow.

The local $V_P$ minimum is laterally continuous, but is interrupted offshore Martinique ($y < -100$), at the projection of the Marathon fracture zone (Fig. 5b). Here the slab surface $V_P$ is slightly higher and there is an increased frequency of small and medium earthquakes[13]. The high $V_P/V_S$ that characterizes the shallow part of the slab extends deeper and a linear high-$V_P/V_S$ anomaly seems to correlate with the projection of the Marathon fracture zone itself (Fig. 5c). These observations are consistent with the presence of a larger proportion of hydrated ultramafic rocks in the slab crust near

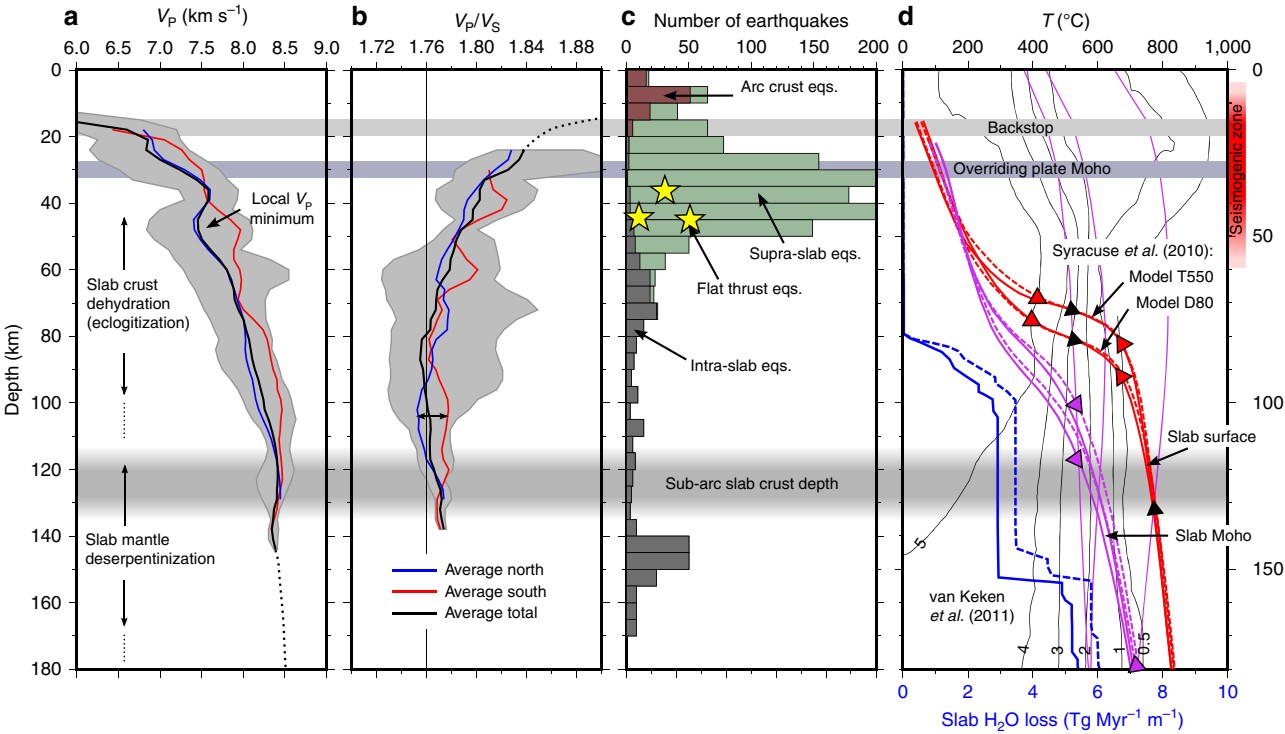

**Figure 6 | Earthquakes distribution and slab dehydration. (a)** Average P-wave velocity and (**b**) $V_P/V_S$ along the slab surface. We show the average for the whole model (black) and the average profiles for the regions north (blue) and south (red) of the Marathon fracture zone. The grey area represents one s.d. from the mean. (**c**) Depth distribution of local earthquakes. We distinguish crustal earthquakes beneath the arc from supra-slab earthquake and intra-slab earthquakes. Yellow stars mark the depth of flat thrust earthquakes located on the plate interface. (**d**) Predicted temperature on the LASZ slab surface (red) and slab Moho (purple)[56] and expected water loss from the slab assuming a MORB crust and serpentinized slab mantle[57] (blue). Dashed lines: southern LASZ; continuous lines: northern LASZ. In the background we show the maximum wt% water content bound in minerals for MORB[56] (thin black lines) and the major dehydration reactions for hydrated peridotite[3] (thin purple lines). Red triangles mark the depth interval of expected water release from dehydration of MORB in the slab crust. Purple triangles mark the expected dehydration of the slab mantle. Black triangles mark the expected dehydration of tectonized slab crust (containing a significant proportion of serpentinized peridotite).

and south of the Marathon fracture zone likely resulting from tectonized accretion at the time of the genesis of the plate near the MAR[10]. A slab crust containing a significant volume of serpentinized peridotite would result in enhanced transport of volatiles and would dehydrate at a deeper level (Fig. 6d), thanks to the deeper stability of hydrated ultramafic phases compared with hydrated MORB[58]. Near the fracture zone the slab mantle may be more hydrated, also thanks to a potentially thinner crust and the presence of fractures facilitating penetration of seawater. Bend faults are also expected to play a role in hydrating the upper mantle, but their effect is likely to be diminished in the south by the thick sediment cover. MCS profiles show evidence for the presence of horst and graben structures in the oceanic basement[41], but south of the Barracuda Ridge bend faults are likely to grow beneath the thick accretionary wedge and do not cause any significant displacement of the seafloor seaward of the deformation front, likely limiting their effectiveness in capturing and transporting water to mantle depths.

The high $V_P/V_S$ in the south may alternatively be related to increased sediment input into the subduction channel, since it correlates with a thicker sediment layer on the incoming plate. $V_P/V_S$ in water-saturated metasediments is likely to be relatively high ($>1.9$) and may be increased further by the effect of anisotropy caused by orientation of fractures parallel to the plate interface due to prolonged shear strain[59]. A thicker subduction channel would also transport additional water into the subduction zone and may thermally insulate the plate from the overlying mantle wedge, delaying the onset of eclogitization and fluid release. Independent evidence for enhanced sediment subduction in the south comes from geochemical studies of lavas, which reveal the

signature of sediment melting in the southern Lesser Antilles[60,61]. Differences in the amount of sediment transport may also be responsible for variations in seismic properties on the shallow part of the plate interface. High-$V_P/V_S$ and low-$V_P$ are found between the Tiburon and Barracuda ridges (Fig. 5c, $Y = 30–90$ km), suggesting that thick sediment packages trapped ahead and/or behind the ridges are being underthrust and transported on the slab top to at least close to the overriding plate Moho. This interpretation is supported by observation on MCS profiles of a thicker subduction channel between the Barracuda and Tiburon ridges that is preserved beyond the backstop[41].

An important consequence of the observation that the slab crust dehydrates at relatively shallow levels ($<100$ km) is that crust-derived fluids are unlikely to be the main drivers for melt generation beneath the arc unless they can be transported $\sim 50$ km arc-ward by the mantle wedge corner flow. We suggest that dehydration of the slab mantle around 120–160 km depth may be the most significant source of the hydrous fluids. This interpretation is supported by observations of enrichment in heavy Mg isotopes in Martinique lavas[62], which suggest influx of serpentinite-derived Mg-rich fluids into the mantle wedge. The location of the deep cluster of seismicity within the slab mantle suggests that these fluids are derived from dehydration of serpentinized slab mantle peridotites and not from serpentinized mantle wedge entrained by the corner flow. At 120–160 km depth temperatures within the slab's upper mantle are expected to reach 500–800 °C (refs 55,57), compatible with dehydration of serpentinite. Upward fluxing of slab mantle-derived fluids through the sediment mélange on the top of the slab would also provide a mechanism for sediment melting, as

the presence of abundant external water has been shown to significantly facilitate melting in high-pressure laboratory experiments[63].

Our new detailed constraints on the physical properties of the subducting slab and their relationship with the distribution of seismicity allow us to re-evaluate the extension of the seismogenic zone, that is, the portion of the plate interface that is locked during interseismic periods and is capable of initiating large earthquakes. Its updip limit is likely to be near the backstop, that is, the landward edge of the accretionary prism[18,40,41]. In our model the backstop coincides with a marked increase in $V_P$ on the plate interface (Figs 5a,b and 6) and may correspond to the depth where pore space collapses and porosity is significantly reduced[50]. Above this depth high pore pressures along the subduction channel are likely to induce stable sliding and in fact very little seismicity has been detected seaward of the backstop despite the extensive deployment of OBSs. However, given the numerous similarities with the Tohoku margin, it is likely that large earthquakes initiated at greater depth may be capable to rupture this region[64].

The downdip limit of the seismogenic zone is most likely deeper than the contact with the overriding plate Moho (which is found at 25–35 km depth) since several $M_w \simeq 5.0$ flat thrust plate interface earthquakes have been observed at over 40 km depth (Figs 4 and 5). The brittle/ductile transition may instead be close to the observed local minimum in $V_P$ at $\sim 50$ km depth. If, as we suggest, the increase in $V_P$ below this local anomaly were related to the blueschist-eclogite transition, it would correspond to temperatures ranging from 350 to 500 °C (ref. 3), which are

predicted to coincide with the onset of ductile behaviour in the subduction channel[65,66]. In addition, fluids released from eclogitization would likely cause overpressure and perhaps the formation of aseismic serpentinites and talc in the mantle wedge, which would also favour stable sliding. This interpretation is supported by the similarity of the deep flat thrust events observed in the Lesser Antilles with well-documented repeating earthquakes in Tohoku[48] that mark the transition zone between stick-slip behaviour and stable shear. On average the seismogenic zone in the study region is predicted to be $\sim 100$ km wide. We note that the most up-to-date rupture area models of the 1839 and 1843 earthquake based on reported damage intensity[15] are inconsistent with the limits of the seismogenic zone determined above and should be revised by a seaward shift of $\sim 20$ km.

We identify several high-$V_P$ patches along the plate interface, which may indicate a strong overriding plate or a relatively dry subduction channel and a locked and potentially loaded megathrust fault. In Tohoku, high $V_P$ on the megathrust correlates with large co-seismic slip during the 2011 great Tohoku earthquake[67], which has been interpreted to indicate long-term locking. On the other hand, high-$V_P$ patches do not always correlate with strong coupling, for example if they correspond to ultramafic blocks[68]. If the relationship between high $V_P$ and coseismic slip can be extended to the LASZ, high $V_P$ patches on the slab surface identified in our study may be expected to produce the largest coseismic slip in future megathrust earthquakes. In the absence of accurate geodetic and seismological constraints on plate coupling, strong ground

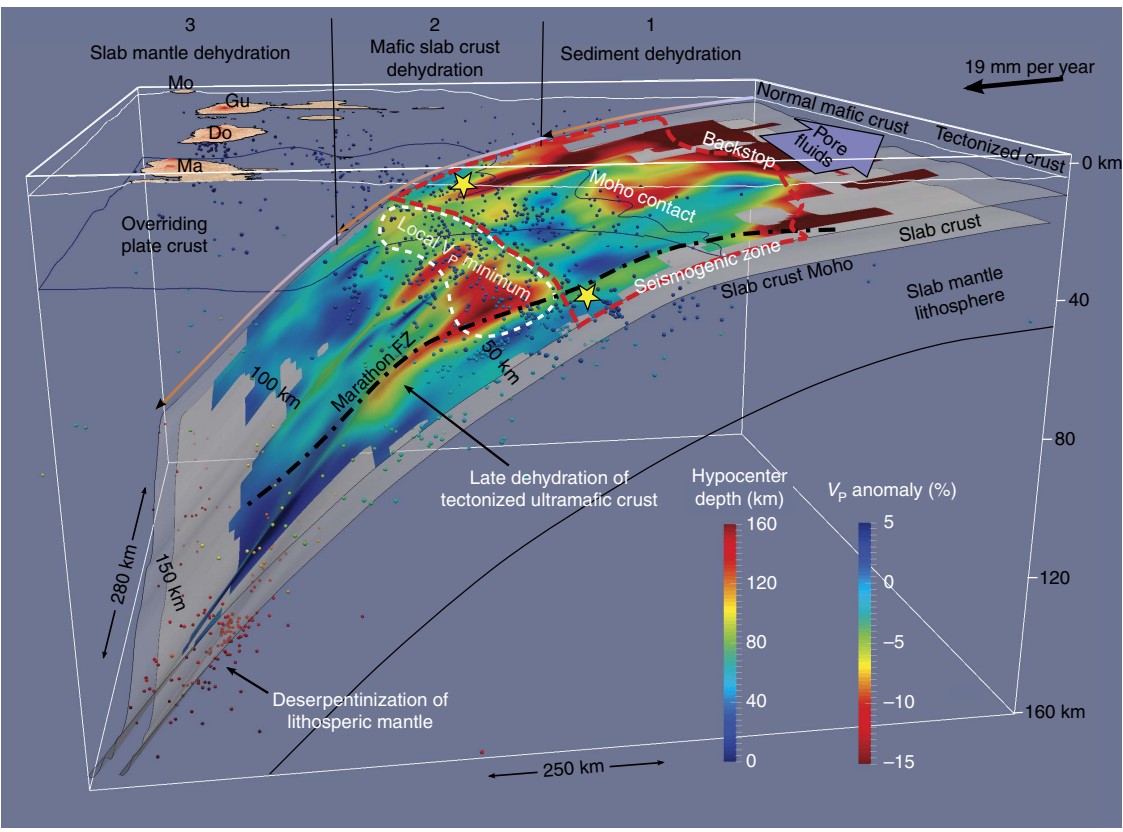

**Figure 7 | Representation of the three stages of slab dehydration.** The $V_P$ anomaly (same as Fig. 5b) is draped over the slab surface. Local earthquakes are marked by coloured dots. The location of the local $V_P$ minimum near 50 km depth is highlighted by a dashed white line. The slab crust Moho is drawn assuming a 7 km crustal thickness. Yellow stars represent the hypocenters of large thrust earthquakes. The proposed extent of the seismogenic zone is marked by a red dashed line. Notice how the mantle wedge seismicity spatially correlates with the local $V_P$ minimum and how intra-slab seismicity is enhanced near the fracture zone. See Fig. 5 caption for further display information. A rotating animation of the slab viewable in 3D is presented in Supplementary Movies 1 and 2.

motion scenarios produced by modelling of earthquake rupture over these patches could help inform regional earthquake hazard mitigation strategies.

On the basis of our tomography model and on the observed seismicity distribution, we suggest that the Atlantic slab undergoes three main phases of dehydration (Fig. 7): (1) compaction and draining of pore fluids from the sediments and upper crust at 5–20 km depth; (2) chemical dehydration of the crust at 40–100 km depth (with a possible peak at 50–80 km depth); and (3) lithospheric mantle deserpentinization at 120–160 km depth. These depth ranges agree with numerical thermo-chemical models[57], which predict surges in water release at 80–100 km depth and 150–160 km depth in the LASZ (Fig. 6d). In the south of the study region, however, this three-stage dehydration model likely breaks down, as a result of a lack in compositional diversity between the crust and mantle associated with a tectonized slab crust. Here eclogitization is likely to be less prevalent and the slab may contain a greater proportion of serpentinized peridotite, carrying an overall larger volume of water to greater depth.

Our results provide new evidence on the relationship between slab dehydration and seismicity, showing that earthquakes in the slab and in the mantle wedge are closely related to hydration and dehydration processes and are controlled by thermal structure, composition and state of stress. The heterogeneous composition and hydration of oceanic plates created by slow and ultraslow accretion, like the Atlantic plate, can result in lateral variations in fluid transport and release during subduction.

## Methods

**Starting model.** The inversion was carried out in a local Cartesian reference frame with origin at 61° W, 16° N and rotated by 24° to align the x axis perpendicular to the strike of the trench. The starting P-wave velocity model was built as a pseudo-1D hanging model (a 3D model built by 'hanging' a single velocity/depth profile below a given surface) referenced to the depth of the crystalline basement, which was constrained by MCS profiles[39–41]. Sediment velocities were assigned to nodes between the seabed and the basement surface, based on previous studies[18,22,30]. The seabed and land surface were defined based on multi-beam bathymetry data[21,69], the GEBCO_2014 Grid, version 20150318 (www.gebco.net) and data from the ASTER Global Digital Elevation Model v2 (ASTER GDEM is a product of NASA and METI, ref. 70). Different crustal velocity/depth profiles were used for the arc, forearc and oceanic plate. Mantle velocities with a constant vertical gradient of $0.005\,s^{-1}$ were added below a smooth Moho discontinuity constrained from receiver functions[71] and 2D wide-angle profiles[20]. No slab was introduced in the starting model. The starting $V_P/V_S$ model had a homogeneous value of 1.76, determined from linear regression of $T_S$ vs. $T_P$ distribution (the Wadati diagram, see Supplementary Fig. 1). This starting pseudo-1D model (Fig. 3a) was preferred to a true minimum 1D model as it represents already a 50% data variance reduction, improving stability and convergence.

**Inversion method.** We used the code Simulps[28], which employs approximate ray tracing and pseudo-bending to solve the forward problem[29] and a combination of parameter separation and damped least squares to tackle the inverse problem. S-wave data were used to calculate S–P traveltimes, which were inverted jointly with P-wave traveltimes for the 3D $V_P$ and $V_P/V_S$ structure as well as hypocenter parameters (coordinates and origin times) and station corrections. The inversion functional is designed to minimize traveltime residuals and a damping term through several iterations. At each iteration the forward and inverse problems are solved and the model and hypocenter parameters are updated. The inversion is stopped when the improvement in data variance is smaller than 1% of the initial data variance. The final RMS residual is 0.26 s (0.24 s for P-traveltimes, 0.39 for S–P traveltimes) and represents a data variance reduction of 83% and 70% for P and P–S traveltimes, respectively, compared to the initial model (Supplementary Table 1; Supplementary Fig. 2).

**Inversion in practice.** We started with an inversion on a sparse grid with 40 km spacing in the along arc direction and 50 km in the across arc direction ($40 \times 50$ km model). The vertical spacing was set to 5 km at the top, 10 km in the shallow mantle wedge and 20 km at the base of the model. Five separate inversions were carried out on different staggered grids shifted by half the grid spacing in the X, Y, Z and X+Y directions. The final models of each inversion were averaged and the average model was then resampled on a grid with horizontal spacing of 20 km and used as a starting model for a more detailed inversion. Again five separate inversions were carried out on staggered grids. The resampling and inversion was repeated once

more for a 15 km grid (Fig. 3). As the final model we selected the median of the five staggered inversions with the $15 \times 15$ km grids, since the median preserves more of the short-wavelength variability than the mean and effectively removes spurious peaks. A horizontal grid spacing of 15 km ensures that data gaps are minimized, while allowing sufficient complexity to model the subducting slab and any strong shallow lateral variations. Some of the smaller scale features in the upper crust, however, cannot be taken into account by our model parameterization and are absorbed by the station corrections.

**Parameter selection.** Three independent damping coefficients need to be defined for $V_P$, $V_P/V_S$ and station corrections. Damping values are often chosen based on analysis of the L-curve, the trade-off between data variance and model variance for single-iteration inversions. We tested a range of damping values for $V_P$ and $V_P/V_S$ and observed the tradeoff between the data and model variance for both single-iteration inversions and multiple-iteration inversions. We use a conservative large damping term for $V_P$ and $V_P/V_S$ in the initial $40 \times 50$ km model inversion and the $20 \times 20$ km model inversion to steer the model towards the global minimum, without introducing too much complexity. For the $15 \times 15$ km model inversion we decreased the damping to the optimal value, determined by analysis of trade-off curves between model and the data variance.

**Ray coverage.** A first qualitative estimate of the resolution of the tomography model can be deduced by investigating the ray coverage (Fig. 2; Supplementary Movies 1 and 2). P-wave ray trajectories from controlled shots cover the crust and upper mantle of the overriding plate. The maximum turning depth is $\sim 35$ km but is usually $< 20$ km. The ray coverage is less dense beneath the arc, particularly between Montserrat and Guadeloupe where few profiles were shot. The ray coverage of the local earthquakes extends much deeper into the mantle wedge and slab, but is sparser at shallow depth where it is mostly controlled by the station distribution. The upper part of the mantle wedge is well covered by rays with a wide range of azimuths and inclinations. In contrast, the slab mantle is crossed by fewer rays that are mostly subvertical, likely resulting in significant smearing. In Supplementary Figs 3c and 4c we plot the derivative weight sum (DWS), a measure of ray density at each grid node weighted by how close the ray comes to the node. The DWS for $V_P$ nodes is everywhere greater than for $V_P/V_S$ nodes because of the greater number of inverted P-wave traveltimes. In the mantle wedge and slab the DWS is generally larger in the south because of a greater density of local earthquakes in this region even after our declustering selection. There are no large ray coverage gaps, except in the shallow crust and at the edges of the model.

**Resolution matrix.** We computed the full resolution matrix, which allows a more detailed evaluation of the information contained in the output model. From the resolution matrix we calculated the spread function (SF, ref. 72), which gives an indication of how peaked the resolution is at each node (Supplementary Figs 3a and 4a), and therefore of the level of smearing. A SF below 2.5 indicates well-resolved areas. In areas with $2.5 < SF < 3.5$ the model may be smoothed over an area larger than the grid spacing. For $SF > 3.5$ the smearing is significant and anomalies are likely to be significantly underestimated[73]. We also plot the 70% contours of the resolution kernel, that is, we trace contours around each node, where the resolution is 70% of the diagonal element, which gives an indication of the amount and direction of smearing (Supplementary Figs 3b and 4b). On the basis of this analysis we expect the crust of the overriding plate, the mantle wedge and the upper 30 km of the slab down to a depth of $\sim 100$ km to be well resolved. The sub-arc mantle wedge and the deeper parts of the slab are likely to suffer from significant subvertical smearing. The $V_P/V_S$ model is less well resolved because of the smaller number of S–P traveltimes and generally larger pick uncertainty. In addition to the regions mentioned above the $V_P/V_S$ model is likely to suffer from vertical smearing in the shallow part of the overriding plate.

**Checkerboard tests.** To further assess the resolution of the model and in particularly the ability to resolve lateral variations in seismic properties we carried out a series of checkerboard tests. We added 5% checkerboard anomalies to our final model. We then calculated predicted traveltimes for all events (shots and local earthquakes) and receivers in the synthetic model and carried out a complete inversion with the synthetic data on the $15 \times 15$ km grid, using the same inversion parameters as in the original inversion. For simplicity we did not carry out five parallel inversions on staggered grids for each checkerboard test.

We used 5% horizontal checkerboard anomalies with cell size of 15 km (a single grid point) and 30 km (two grid points). We tested anomalies placed at depths of 6–25 km, 35–70 km and 70–100 km. For $V_P$ the tests show that in the upper crust even very small anomalies can be recovered over most of the study area. The larger anomalies are reasonably well recovered at over 80 km depth (Supplementary Fig. 5). The fidelity of the recovered pattern is better in the south of the study area particularly at depths greater than 20 km. The 15 km $V_P/V_S$ anomalies can be recovered only in limited regions and at shallow depth. The 30 km $V_P/V_S$ anomalies are well recovered to at least 80 km depth (Supplementary Fig. 6).

**Slab anomaly tests**. To specifically assess the robustness of the slab anomaly we tested the ability of the inversion and data distribution to recover a synthetic slab anomaly including a dipping low-$V_P$ layer corresponding to the slab crust. We started with our final model and removed all structure beneath the Moho, we then added a 15-km thick slab-shaped $-8\%$ $V_P$ anomaly transitioning to a $+4\%$ anomaly at 90 km depth (Supplementary Fig. 7). We used a 15 km thick anomaly to represent the combined effect of the slab crust, the sediment cover and a partially serpentinized slab upper mantle. In the $V_P/V_S$ model we added a positive anomaly, with $V_P/V_S$ increased from the background value of 1.76 to 1.82. Following the same workflow used in the checkerboard tests, we calculated synthetic traveltimes through this model and performed a full inversion on the 15x15 km grid with the synthetic data as input, to test how well the anomaly can be recovered. We find that the shape of the anomaly is well recovered, with an amplitude of 50–70% of the input anomaly, and that its termination depth can be correctly determined to within 10 km. We carried out a second test with a deeper anomaly reaching 120 km depth (Supplementary Fig. 8). In this case the termination depth cannot be determined with confidence.

We also carried out checkerboard tests for the top of the slab. We added 5% anomalies to the top of the slab in our final model and followed the same procedure described above to calculate the synthetic data and run the inversion to recover the input anomaly. We tested different anomaly distributions including trench-parallel and trench perpendicular bands and square checkerboards, with cell sizes of 30 and 60 km, for both $V_P$ (Supplementary Fig. 9) and $V_P/V_S$ (Supplementary Fig. 10). In the vertical direction the anomalies start at the slab top surface (as identified in this study) and extend to 15 km beneath the slab top.

The anomalies are relatively well recovered within the area identified as being well resolved based on the study of the resolution matrix. As expected, the linear anomalies are better recovered than the square anomalies. The amplitudes of the $V_P$ anomalies are well recovered (50–80%). The recovered $V_P/V_S$ anomalies have lower amplitudes, suggesting that some of the $V_P/V_S$ structure may be attenuated in our final model.

**Anisotropy**. We do not account for anisotropic effects, which are recognized to cause artifacts in tomography models when not accounted for during inversion[74]. The largest effect is expected for the sub-arc and back-arc mantle wedge and sub-slab asthenospheric mantle, where mantle flow causes alignment of olivine crystals. Recent constraints on anisotropy in the LASZ have been obtained from shear wave splitting measurements[75], and indicate a relatively isotropic mantle wedge. We strive to minimize the impact of unaccounted anisotropy on our result by specifically concentrating our interpretation on the properties of the slab and shallow mantle wedge, and on the longer wavelength structures.

**Data availability**. The traveltime database assembled as part of this study and coming from the SISMANTILLES 1 and 2, OBSAntilles, OBSISMER, TRAIL and SEA-CALIPSO experiments are available from the corresponding authors on request. Requests for the traveltimes from EW9803 cruise should be addressed to the cruise's PIs. The final $V_P$ and $V_P/V_S$ models will be shared after publication on the first author's Institution website and on ResearchGate.

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

## Acknowledgements

This research was supported by an AXA Research Fund Postdoctoral Fellowship and the OBSIVA Project funded by the Prix de la Fondation Simone et Cino del Duca of the French Academy of Science. This work has been initially funded by the European Union FP6 NEST (New and Emerging Science and Technology)—INSIGHT programme, under project 'THALES WAS RIGHT' no. 029080. We are grateful to the Montserrat Volcano Observatory for sharing their data and to the PIs, scientist, students and technical staff that designed and participated to the expeditions EW9803, SEA-CALIPSO, Sismantilles 1 and 2, TRAIL, OBSAntilles and OBSISMER, in particular to Ernst Flueh, Alfred Hirn and Gail Christeson. We have benefited from the work of Mario Ruiz who extracted and initially relocated part of the local earthquakes database. We thank Claude Jaupart, head of IPGP between 2010 and 2015, for his support to the project. We thank Donna Eberhart-Phillips and Cliff Thurber for making the tomography code Simulps public. All figures where generated using the Generic Mapping Tools or Paraview and edited with InkScape.

## Author contributions

M.P. put together the traveltime database, picked ∼50% of the active source traveltimes, carried out the tomographic inversion and wrote the manuscript. M.L., A.G. and P.C. provided access to data and contributed to the tomography and the manuscript. M.S. installed and serviced temporary land seismometers and identified, picked and initially relocated the majority of the local earthquakes. G.B. and M.E. processed part of the active source dataset as part of their doctoral theses at IPGP and Géoazur, respectively. H.K. provided the traveltimes of the northern 2D across-arc profile and contributed to the discussion and interpretation.

## Additional information

**Competing interests:** The authors declare no competing financial interests.

