## [Peer Review File · Nature Communications]

Reviewers' Comments:

Reviewer #1 (Remarks to the Author)

In this article, Paulatto and co-authors present a synthesis of seismological data acquired over several geophysical survey campaigns in the Lesser Antilles subduction zone.

The Vp and Vp/Vs anomalies and seismicity in the slab and overlying mantle wedge are interpreted in terms of dehydration reactions within the oceanic lithosphere and of the large influx of sediments/tectonized lithosphere toward the southern part of the studied region. The data appear to show that arc-volcanism is related with fluids expelled from the slab due to mantle dehydration reactions. Implications for megathrust properties and interplate coupling are also discussed.

I believe this is a comprehensive study of the 3D metamorphic, magmatic and mechanical (i.e., seismogenic) structure/behaviour of a subduction zone, which is attractive to a broad audience. I would be happy to see these data published in such a high-profile journal.

I have only minor comments, mostly related to the presentation of the results, some additional discussion and few slightly speculative statements:

- the choice of the Vp (and also Vp/Vs) anomalies colour scale is a bit confusing, as fast and slow anomalies are usually shown with blues and red colours, respectively, and not vice versa.
- Figure 1: to which profile do the orange lines correspond to? Also, I cannot see the green lines of the EW9803 profiles.
- Figure 2: what are the inverted red and grey triangles in the Vp images?
- Figure 4: blurry.
- Line 132: mantle wedge earthquakes in profiles 3 and 4 seem to be associated with high Vp and normal Vp/Vs. In general the interpretation of this type of seismicity is a bit weak. What are the potential phase transitions that can cause a volume change? Are pyroxenite and pore fluid overpressure consistent with observations in profile 3 and 4?
 - Lines 163-170: Is there any offshore evidence of the hydration state of the plate before entering the deformation front? Eventually, the statement about a higher hydration of the Marathon fracture zone could be made more robust.
 - Lines 189-193: this part is a bit speculative, as sediments could explain the observations (lines 172 – 181). Again, if available, it would be better to constrain the statement about the presence of tectonized lithosphere with some offshore geophysical evidence (gravity, seismic anomalies, for example?).
 - Is there any evidence of bending-related normal faulting offshore of the deformation front (in the bathymetry maps or in the seismicity record)? Or is the bending more active beneath the accretionary prism as indicated by intraplate seismicity?. Please discuss more this point, as from the introductory part it looks like that you assume that plate hydration is related only to tectonization of the oceanic crust at slow spreading ridges and near fractures.
 - How the Vp/Vs ratios compare with those of Eberhart-Phillips et al., 2013, PEPI, that have analogously tried to interpret these type of data and intermediate-depth seismicity in terms of slab dehydration along the Hikurangi subduction zone?

Manuele Faccenda

Reviewer #2 (Remarks to the Author)

Paulatto and co-authors present a comprehensive 3D Vp and Vp/Vs model of the Lesser Antilles subduction one based on a remarkable joint analysis of active source data and locally recorded earthquakes. This model exhibits very interesting along-strike and down-dip changes in large-scale

velocity structure that the authors interpret in terms of hydration and dehydration of the crust and upper mantle of the subducting oceanic lithosphere. This subduction zone is relatively unique in that slow-spreading crust produced at the Mid-Atlantic ridge is subducting here (in contrast to the fast-spreading crust subducting at most other subduction zones around the world).

The most compelling conclusions of the paper are the large-scale patterns of dehydration of the crust and upper mantle that they infer from their models of V_p and V_p/V_s . Their new constraints on the depth at which the oceanic crust and upper mantle dehydrate with respect to earthquakes and the volcanic arc are important and new, and warrant publication in Nature Communications.

On the other hand, I found many of the smaller scale features that they discuss and interpret to be much less convincing. Their interpretation of features related to subducted fracture zones, megathrust fault properties and sediments seem tenuous based on their model resolution (and even their 3-5 km grid spacing!). As impressive as their model is for examining large-scale structures, it does not resolve the scales relevant for assessing properties of the megathrust fault relevant for slip behavior (e.g., pore-pressure, detailed sediment properties on the megathrust fault, etc), for example Lines 208-242 or changes in properties of the sediments and upper crust (Lines 185-186). The resolution tests assess structures at the scale of 30-60 km, but the scale lengths needed to consider megathrust properties are an order of magnitude smaller. Discussing these points, which are not well supported by this model, dilutes the impact of their findings about the large-scale hydration story, which can be addressed with this model. As a result, there is a less space to actually develop those.

In summary, the combination of many active source experiments and local earthquakes to create a single model of subduction zone structure to large depths is important. These constraints are in a global end-member of Earth's subduction zones – slow-spreading crust is subducting here, which makes this an interesting and important place to study. They observe large-scale variations in hydration and dehydration with respect to seismicity and arc magmatism that are very important – these aspects are worthy of publication in Nature Communications. However, they also discuss many other smaller scale features that I do not think are supported by the resolution of their model (megathrust fault properties). I think the paper should be rewritten to focus on the large-scale hydration story and omit the smaller scale features.

Detailed comments:

Line 20 – Its not really the seismic inversion methods, alone, that can constrain fluids – it's the actual data.

Line 54 – Slow convergence rate does not contribute to a colder subduction zone (faster convergence favors colder subduction zones). Rephrase this sentence to make it clear that the reason this subduction zone is cold is because of the exceptionally old plate.

Line 165 – Why would higher V_p be associated with the projection of the Marathon FZ? That would not be consistent with hydration/deformation. I guess its possible that lightly serpentized peridotite would be faster than lower crust gabbro in some cases, but then would you have enough hydration to explain the high V_p/V_s ?

Line 170 – Some justification or reference is needed for the statement that a more hydrated slab would transport fluids to greater depths.

Lines 172-181 – By the time subducted sediments reach depths of the Moho of the overriding plate, they would have been metamorphosed and transformed in crystalline rocks. Its not clear to me that they would have properties that are distinct from the overriding crust and could be resolved by this coarse model. This point needs to be much better justified or omitted.

Lines 185-186 – I did not see any strong evidence in this paper that constraints can be placed on

the evolution of the sediments or upper crust from this model. These layers would be too small to be resolved by this model.

Line 209 – “Our new detailed constraints on the physical properties of the megathrust fault” This study does not place any constraints on the detailed properties of the megathrust fault. It does provide constraints on the slab hydration and overall geometry. This needs to be rephrased.

I think that the writing could be improved to be more clear and succinct.

Figure 1 – what experiment is represented by orange lines and dots? This information is not in the caption.

Figure 2 – I find the Vp anomaly and Vp/Vs color scales reversed from what I’m accustomed to. Usually red is used to indicate slow velocities and high Vp/Vs ratios. Also, choose a better color scale for Vp. Nearly the entire velocity range of interest is red in this figure, so its difficult to see the features described in the text.

Reviewer #3 (Remarks to the Author)

General comments:

This paper combines data from several seismic experiments and deployments in the Lesser Antilles arc to produce a high resolution seismic tomography of the region. These results are interpreted in terms of the hydrated structure, and dehydration processes that occur within the Lesser Antilles arc. The results are clearly exciting, and certainly represent an increase in resolution in the Lesser Antilles arc, and therefore have the potential to offer new constraints on dehydration processes in this atypical and complex setting.

However, limited meaningful discussion of the extent to which features such as a localised Vp decrease can be imaged, or the lack of a hydrated low velocity layer at depth can be inferred, mean that several of the papers claims are not fully justified. The text also lacks polish and focus. While the introductory paragraphs set the scene for why these results are potentially interesting well, these points are not then picked up with the same sharpness in the discussion section of the paper.

This studies requires a greater level of resolution testing in order to justify the interpretation of fine scale velocity structures that the authors claim to have imaged. The manuscript also needs significant revision in the discussion parts of the manuscript, to make the core original claims and findings of the work clearer to the reader. However if these changes can be made effectively I think this work has the potential to inform a new level of detailed understanding of how a subducting plate dehydrates.

Major revisions:

1. Resolving the crustal low velocity layer

The authors propose that a low velocity oceanic layer, interpreted as hydrated oceanic crust is imaged at depths of up to 100 km by this travel time tomography. The P-wave velocity of the proposed oceanic crust is shown with increasing depth, and details of the velocity are interpreted. The authors claim that this low velocity layer ‘disappears’ at below 70-100 km depth, and attribute this to the eclogite transformation. This is a bold claim, and I am far from convinced that a travel time tomography would be capable of imaging a crustal low velocity layer at these depths, especially given the decreased ray path coverage at depth. It is notable from supplementary figure 5 that the Vp spread function increases to above 2 at approximately 70-100 km depth, indicating that the resolution in turn is decreasing precisely at these depths.

If the low velocity layer is to be interpreted in any way at these depths, the authors must show specific resolution tests establishing to what extent a crustal low velocity layer could be resolved

by this travel time tomography. These resolution tests may also allow the authors to make some comment on the thickness of the proposed crustal low velocity layer at depth, which at present is missing. The checkerboard tests shown in supplementary figure 7 seem to suggest that there is little resolution below 100 km depth. More generally more rigorous examination of the models resolution is needed to test the validity of the authors interpretation, as discussed in more detail in point 2.

2. Checker board tests

The only resolution tests that are currently shown are checkerboard tests along the slab interface (supplementary figure 7). While these are of course important to test the assertion that along strike variations can be imaged, there are several earlier tests of resolution that are needed, e.g. to justify the claim that a low P-wave velocity crustal layer is imaged to 70-100 km depth. The supplementary material should at least show a checkerboard test for a vertical slice on one (or probably all) of the vertical slices where the reduced crustal P-wave velocity is proposed. To really assure the reader that this structure can be resolved I would strongly recommend that you also perform resolution tests with a low velocity layer of the type described in the text (7 km thick and 5-10 % slower than the surrounding material), to check the depth to which your travel time tomography is able to resolve such features. Otherwise your argument that you can not only image this low velocity layer, but also resolve along strike changes in it, is left largely unsubstantiated.

In the checker board tests that are shown please clarify whether the checkerboards are limited to the proposed low velocity crust (e.g. 7 km in thickness), or are projected throughout the model area?

3. Local Vp minimum

This is a very specific and fine scale low velocity structure to interpret from a travel time tomography. A greater set of checkerboard tests (as described above) may go some way to assuring the reader that these features are resolved. But if you really want to test it, you could again test the model with a block model. Otherwise much of your discussion is resting on what untested and potentially unstable observations.

Minor revisions:

Line 34: You mention that outer rise faults have a potentially significant role to play, but do not pick up on this point again. Do these structures have any role in the Lesser Antilles, or is there any evidence from your study?

Line 53: "a cold subduction" this is poorly worded. More precisely it is a subduction zone with a low thermal parameter.

Line 59-61: It is unclear what you mean here. You mention two large thrust events. Are you suggesting that these compose of one sequence? This seems unlikely. Better to simply refer to the two historical events.

Line 70-71: These statements need more detail or appropriate referencing. Which two large events? What evidence is there that the long term volcanic activity is more vigorous?

Line 75: "one of the largest combined traveltimes databases." This statement needs conditioning. Is this one of the largest travel time data sets in any geophysical study, or in a subduction zone setting, or just in the Lesser Antilles arc? Currently this is unclear.

Line 84-85: This is unclear to me. What is a hanging 1D model? Do you mean that initial 1D model with a variable upper crust? Please clarify here and in the supporting info, and define the term 'hanging 1D model' with the non-seismologist reader in mind.

Line 107: I am confused why you are sighting the Abers (2005) guided wave paper here, since you are not referring directly to this work. It would be better to more clearly state that a crustal low velocity layer is a phenomena noted in other subduction zones through guided wave studies as you do below.

Line 109: "Our inversion provides one of the clearest images of a subducting slab crust". Perhaps best making sure you say 'tomographic images' here as there are many far clearer receiver function images available in other arcs. Also consider broadening you referencing of other high resolution tomographic studies of the low velocities in subduction zones, particularly Zhang et al (2004, Geology) in Japan.

Line 110-111: Please make it clear to the reader that all of the studies you refer to were conducted in other arcs. Your choice of papers imaging subducted low velocity crustal material in seems a little random as well.

Waveguide behavior – I am confused why you do not refer to Abers (2005) here, as you have already mentioned it earlier in the text. You should probably refer to Abers (2000) as this is the classic guided wave paper. There are also other more recent guided wave papers that consider subducting crust of similar age (e.g. Garth & Rietbrock 2014, GJI).

There are many many receiver function paper showing low velocity structure. So perhaps just refer to one or two significant papers. Yuan et al (200) is an early example, and Abers (2013) gives an overview of many arcs for example.

Line 134-136: This statement isn't supported by the paper – can you refer to a study that suggests this or justify this statement otherwise?

Line 138: I would strongly suggest referring to this as the Wadati-Benioff zone throughout the paper, as this is the most widely accepted and correct terminology.

Line 142: appears distributed > appears to be distributed.

Line 149-150: I must reiterate my strong concerns that this increase in velocity may be simply because you cannot resolve any low velocity feature that may be there.

Line 150-151: This is a very general statement. Given the thermal model you refer to later in the text, can you be more specific? Are the pressure and temperature conditions expected to be in the eclogite stability field?

Line 214-217: This is very similar to the high pore fluid pressure at the nose of the Japanese trench which of course supported slip during the Tohoku earthquake. Perhaps change the focus of this statement.

Figure 3: perhaps also show the mantle wedge seismicity here, as you link it with the proposed low velocity area in the text.

Figure 4: The Vp reduction is shown for the North of the area, when I believe it is seen in the south of the area (figure 3).

Reviewers' comments:

Reviewer #1 (Remarks to the Author):

In this article, Paulatto and co-authors present a synthesis of seismological data acquired over several geophysical survey campaigns in the Lesser Antilles subduction zone. The V_p and V_p/V_s anomalies and seismicity in the slab and overlying mantle wedge are interpreted in terms of dehydration reactions within the oceanic lithosphere and of the large influx of sediments/tectonized lithosphere toward the southern part of the studied region. The data appear to show that arc-volcanism is related with fluids expelled from the slab due to mantle dehydration reactions. Implications for megathrust properties and interplate coupling are also discussed.

I believe this is a comprehensive study of the 3D metamorphic, magmatic and mechanical (i.e., seismogenic) structure/behaviour of a subduction zone, which is attractive to a broad audience. I would be happy to see these data published in such a high-profile journal.

I have only minor comments, mostly related to the presentation of the results, some additional discussion and few slightly speculative statements:

- *the choice of the V_p (and also V_p/V_s) anomalies colour scale is a bit confusing, as fast and slow anomalies are usually shown with blues and red colours, respectively, and not vice versa.*

A: We have changed the color scale for the seismic velocity anomaly. We tweaked the color scale for V_p to improve contrast. We left the V_p/V_s color scale unchanged since we like to show high V_p/V_s in blue. We find this more intuitive since high V_p/V_s can be considered in some cases as an indicator of high water content.

• *Figure 1: to which profile do the orange lines correspond to? Also, I cannot see the green lines of the EW9803 profiles.*

A: An earlier version of the figure had green instead of orange lines for cruise EW9803 and the text had not been changed. Now fixed.

• *Figure 2: what are the inverted red and grey triangles in the V_p images?*

A: These indicate the location of axis of the active arc (red) and of the ancient arc (gray). Added in the caption

• *Figure 4: blurry.*

A: This was due to the way Microsoft Word deals with figures. It should be improved in the new version. High resolution PDF versions of each figure will be provided in the final submission.

• *Line 132: mantle wedge earthquakes in profiles 3 and 4 seem to be associated with high V_p and normal V_p/V_s . In general the interpretation of this type of seismicity is a bit weak. What are the potential phase transitions that can cause a volume change? Are pyroxenite and pore fluid overpressure consistent with observations in profile 3 and 4?*

A: Pyroxenite is a possibility and has been suggested before for this area. Laigle et al. (2013, Tectonophysics) suggest that mantle wedge seismicity may be indeed an indication of pyroxenite in the mantle wedge and suggest the presence of pyroxenite may be linked to the oceanic plateau origin of the Caribbean plate. We have now expanded this discussion in the text.

• *Lines 163-170: Is there any offshore evidence of the hydration state of the plate before entering the deformation front? Eventually, the statement about a higher hydration of the Marathon fracture zone could be made more robust.*

A: Not yet, however a seismic experiment (Voilà project, J.C. Collier, T. Henstock, A. Rietbrock) is scheduled for May 2017 to investigate precisely this issue.

• *Lines 189-193: this part is a bit speculative, as sediments could explain the observations (lines 172 – 181). Again, if available, it would be better to constrain the statement about the presence of tectonized lithosphere with some offshore geophysical evidence (gravity, seismic anomalies, for example?).*

A: There is indisputable evidence that the crust being accreted today at the section of the Mid-Atlantic ridge is tectonized (Smith et al., 2008, G-cubed). We have now added a reference. No strong geophysical constraints exist for the crust currently subducting, but this will be tested soon by the new experiment mentioned above. The gravity signature on the incoming plate is dominated by the presence of the Tiburon and Barracuda ridges and is damped by the sediment cover, but is otherwise compatible with a tectonized seafloor (lots of fracture zones, medium wavelength variations).

• *Is there any evidence of bending-related normal faulting offshore of the deformation front (in the bathymetry maps or in the seismicity record)? Or is the bending more active beneath the accretionary prism as indicated by intraplate seismicity? Please discuss more this point,*

as from the introductive part it looks like that you assume that plate hydration is related only to tectonitization of the oceanic crust at slow spreading ridges and near fractures.

A: Bend faults are likely to exist in the area as MCS profiles showed evidence of displacement of the oceanic basement beneath the accretionary prism (Laigle et al., 2013, Tectonophysics). Since bend faulting happens landward of the deformation front (beneath the prism), these faults do not break the seabed, which may limit water penetration. Stein et al. (1982) documented a large normal fault earthquake ($M_s=7.5$, 25/12/1969), which they attribute to plate bending, and localized it beneath the accretionary prism. We think that hydration by water penetration through bend faults plays a role but is likely limited compared to other regions.

Further north, where sediment cover is thin and the prism is much reduced in width and thickness, bend faults break the seabed seaward of the deformation front and are observed as evident scarps in multi-beam bathymetry data and as negative polarity reflectors in multi-channel seismic data (data presented at AGU Fall meeting 2016).

A tectonized crust is likely to be more hydrated on average because of several mechanisms: 1: Detachment faulting near the ridge axis; 2. Water seepage and low-T metamorphism near fracture zones; 3. Greater capacity to incorporate water.

• *How the V_p/V_s ratios compare with those of Eberhart-Phillips et al., 2013, PEPI, that have analogously tried to interpret these type of data and intermediate-depth seismicity in terms of slab dehydration along the Hikurangi subduction zone?*

A: Eberhart-Phillips et al find low V_p/V_s in the slab mantle and high V_p/V_s in the slab crust and in mantle wedge above the slab (only where the slab is still relatively shallow, < 50 km depth). The actual values are similar to what we find in the Lesser Antilles, around 1.80. Hikurangi is a cold subduction zone like the Antilles so we can expect many aspects to be similar. It is interesting that we find significant lateral variation in the Antilles, while on Hikurangi the variations are smaller. This supports our interpretation that the slow-spread lithosphere of the Lesser Antilles slab is laterally heterogeneous in water content and composition. We have added a reference to *Eberhart-Phillips et al., 2013, PEPI* in the text.

Manuele Faccenda

A: Thanks for your constructive comments

Reviewer #2 (Remarks to the Author):

Paulatto and co-authors present a comprehensive 3D V_p and V_p/V_s model of the Lesser Antilles subduction zone based on a remarkable joint analysis of active source data and locally recorded earthquakes. This model exhibits very interesting along-strike and down-dip changes in large-scale velocity structure that the authors interpret in terms of hydration and dehydration of the crust and upper mantle of the subducting oceanic lithosphere. This subduction zone is relatively unique in that slow-spreading crust produced at the Mid-Atlantic ridge is subducting here (in contrast to the fast-spreading crust subducting at most other subduction zones around the world).

*The most compelling conclusions of the paper are the large-scale patterns of dehydration of the crust and upper mantle that they infer from their models of V_p and V_p/V_s . Their new constraints on the depth at which the oceanic crust and upper mantle dehydrate with respect to earthquakes and the volcanic arc are important and new, and warrant publication in *Nature Communications*.*

On the other hand, I found many of the smaller scale features that they discuss and interpret to be much less convincing. Their interpretation of features related to subducted fracture zones, megathrust fault properties and sediments seem tenuous based on their model resolution (and even their 3-5 km grid spacing!).

As impressive as their model is for examining large-scale structures, it does not resolve the scales relevant for assessing properties of the megathrust fault relevant for slip behavior (e.g., pore-pressure, detailed sediment properties on the megathrust fault, etc), for example Lines 208-242 or changes in properties of the sediments and upper crust (Lines 185-186). The resolution tests assess structures at the scale of 30-60 km, but the scale lengths needed to consider megathrust properties are an order of magnitude smaller. Discussing these points, which are not well supported by this model, dilutes the impact of their findings about the large-scale hydration story, which can be addressed with this model. As a result, there is a less space to actually develop those.

In summary, the combination of many active source experiments and local earthquakes to create a single model of subduction zone structure to large depths is important. These constraints are in a global end-member of Earth's subduction zones – slow-spreading crust is subducting here, which makes this an interesting and important place to study. They observe large-scale variations in hydration and dehydration with respect to seismicity and arc magmatism that are very important – these aspects are worthy of publication in Nature Communications. However, they also discuss many other smaller scale features that I do not think are supported by the resolution of their model (megathrust fault properties). I think the paper should be rewritten to focus on the large-scale hydration story and omit the smaller scale features.

A: We integrate as many observations as possible to build a complete and compelling story. We make several new observations that allow us to say something new and important about seismogenesis in this region. In particular we constrain for the first time the approximate location of the slab/Moho contact along the whole segment; the extent of the cold nose of the mantle wedge, and the accurate location of mantle wedge seismicity. We identify the depth of eclogitization and the clusters of intra-slab seismicity, which give us new constraints on the thermal structure and likely depth of fluid escape. Regarding the shallow dewatering, we see an initial steep gradient in Vp along the slab top that becomes gentler beyond the contact with the overriding plate Moho. It is true that we cannot say whether this is the result of variations in the subduction channel or in the oceanic crust, because of limited vertical resolution, therefore we interpret it as evidence for a general reduction in porosity. It is well known that sediments in the subduction channel undergo compaction and lose most of the pore water in the first 20 km [e.g. Calahorrano et al., 2008; Saffer and Tobin, 2011]. We have now included these additional references.

Detailed comments:

Line 20 – Its not really the seismic inversion methods, alone, that can constrain fluids – it's the actual data.

A: Noted and edited.

Line 54 – Slow convergence rate does not contribute to a colder subduction zone (faster convergence favors colder subduction zones). Rephrase this sentence to make it clear that the

reason this subduction zone is cold is because of the exceptionally old plate.

A: True, this was unclear, now clarified

Line 165 – Why would higher V_p be associated with the projection of the Marathon FZ? That would not be consistent with hydration/deformation. I guess it is possible that lightly serpentinized peridotite would be faster than lower crust gabbro in some cases, but then would you have enough hydration to explain the high V_p/V_s ?

A: Here we are assuming that the linear V_p minimum at 50 km depth is due to high pore pressures from fluids released by blueschist/eclogite transition. At the Marathon fracture zone this low- V_p anomaly is interrupted perhaps because there isn't such a sudden and voluminous water release because of an absence or reduction in thickness of the volcanic layer. The fact that the V_p/V_s ratio remains relatively high at greater depth supports the idea that the tectonized crust near the fracture zone may carry water deeper.

Line 170 – Some justification or reference is needed for the statement that a more hydrated slab would transport fluids to greater depths.

A: Here we rely on the fact that hydrated peridotites are stable at greater pressures than hydrated gabbros and would therefore release water at greater depth. We have now expanded this explanation and added a reference to Ulmer, P. & Trommsdorff, V. Serpentine stability to mantle depths and subduction-related magmatism. *Science* 256, 858–861 (1995).

Lines 172-181 – By the time subducted sediments reach depths of the Moho of the overriding plate, they would have been metamorphosed and transformed in crystalline rocks. Its not clear to me that they would have properties that are distinct from the overriding crust and could be resolved by this coarse model. This point needs to be much better justified or omitted.

A: Carbonate metasediments like marble tend to have high V_p/V_s [e.g. Wang et al., 2012, GRL]. Other possible mechanisms for increasing V_p/V_s are increased pore pressures, or high pore pressure over a thicker layer. A thicker sediment layer might also insulate the plate, delaying onset of eclogitization and fluid release. We have added more discussion and an additional reference in the text.

Lines 185-186 – I did not see any strong evidence in this paper that constraints can be placed on the evolution of the sediments or upper crust from this model. These layers would be too small to be resolved by this model.

A: It's true that we cannot say much on the thickness of subducted sediments and their fate during subduction from our seismic tomography. However as noted above we can rely on previous observations from different datasets and general theories to complete the picture. We have added some text and references to clarify.

Line 209 – “Our new detailed constraints on the physical properties of the megathrust fault” This study does not place any constraints on the detailed properties of the megathrust fault. It does provide constraints on the slab hydration and overall geometry. This needs to be rephrased.

A: Rephrased for accuracy

I think that the writing could be improved to be more clear and succinct.

A: Are you referring to this paragraph or to the whole manuscript? We have strived to write as clearly and succinctly as possible. English is a second language for all authors (not an excuse).

Figure 1 – what experiment is represented by orange lines and dots? This information is not in the caption.

A: This was a mistake due to a previous version of the image having green lines instead of orange lines for experiment EW9803. Now corrected

Figure 2 – I find the V_p anomaly and V_p/V_s color scales reversed from what I'm accustomed to. Usually red is used to indicate slow velocities and high V_p/V_s ratios. Also, choose a better color scale for V_p . Nearly the entire velocity range of interest is red in this figure, so its difficult to see the features described in the text.

A: The convention blue=fast=cool, red=slow=hot is useful in mantle tomography but not so useful when the crust is included in the model (the crust and particularly the sediments are cooler than anything else but would be red). To make the variations stand out more we have stretched the color palette at low V_p . We have reversed the V_p anomaly color palette, but have kept the V_p/V_s color palette as it was. With this convention high V_p/V_s is blue and is associated with high water content.

Reviewer #3 (Remarks to the Author):

General comments:

This paper combines data from several seismic experiments and deployments in the Lesser Antilles arc to produce a high resolution seismic tomography of the region. These results are interpreted in terms of the hydrated structure, and dehydration processes that occur within the Lesser Antilles arc. The results are clearly exciting, and certainly represent an increase in resolution in the Lesser Antilles arc, and therefore have the potential to offer new constraints on dehydration processes in this atypical and complex setting.

However, limited meaningful discussion of the extent to which features such as a localised V_p decrease can be imaged, or the lack of a hydrated low velocity layer at depth can be inferred, mean that several of the papers claims are not fully justified. The text also lacks polish and focus. While the introductory paragraphs set the scene for why these results are potentially interesting well, these points are not then picked up with the same sharpness in the discussion section of the paper.

This studies requires a greater level of resolution testing in order to justify the interpretation of fine scale velocity structures that the authors claim to have imaged. The manuscript also needs significant revision in the discussion parts of the manuscript, to make the core original claims and findings of the work clearer to the reader. However if these changes can be made effectively I think this work has the potential to inform a new level of detailed understanding of how a subducting plate dehydrates.

Major revisions:

1. Resolving the crustal low velocity layer

The authors propose that a low velocity oceanic layer, interpreted as hydrated oceanic crust is imaged at depths of up to 100 km by this travel time tomography. The P-wave velocity of the proposed oceanic crust is shown with increasing depth, and details of the velocity are interpreted.

The authors claim that this low velocity layer 'disappears' at below 70-100 km depth, and

attribute this to the eclogite transformation. This is a bold claim, and I am far from convinced that a travel time tomography would be capable of imaging a crustal low velocity layer at these depths, especially given the decreased ray path coverage at depth. It is notable from supplementary figure 5 that the V_p spread function increases to above 2 at approximately 70-100 km depth, indicating that the resolution in turn is decreasing precisely at these depths.

If the low velocity layer is to be interpreted in any way at these depths, the authors must show specific resolution tests establishing to what extent a crustal low velocity layer could be resolved by this travel time tomography. These resolution tests may also allow the authors to make some comment on the thickness of the proposed crustal low velocity layer at depth, which at present is missing. The checkerboard tests shown in supplementary figure 7 seem to suggest that there is little resolution below 100 km depth. More generally more rigorous examination of the models resolution is needed to test the validity of the authors interpretation, as discussed in more detail in point 2.

A: We have carried out more extensive resolution tests including slab anomaly recovery tests, which we discuss below and in the text and show that we can resolve the termination of the crustal low velocity layer at least in the Y range -130 to +30 (from Martinique to Southern Guadeloupe). The tomography of course is inherently smooth and therefore the recovered anomaly is reduced. For this reason, we do not discuss in detail the magnitude of the anomaly and we don't venture into estimates of serpentinization or porosity.

2. Checker board tests

The only resolution tests that are currently shown are checkerboard tests along the slab interface (supplementary figure 7). While these are of course important to test the assertion that along strike variations can be imaged, there are several earlier tests of resolution that are needed, e.g. to justify the claim that a low P-wave velocity crustal layer is imaged to 70-100 km depth.

The supplementary material should at least show a checkerboard test for a vertical slice on one (or probably all) of the vertical slices where the reduced crustal P-wave velocity is proposed.

A: We had now included new figures showing several checkerboard tests. Extra explanatory text is included in the method section.

To really assure the reader that this structure can be resolved I would strongly recommend that you also perform resolution tests with a low velocity layer of the type described in the text (7 km thick and 5-10 % slower than the surrounding material), to check the depth to which your travel time tomography is able to resolve such features. Otherwise your argument that you can not only image this low velocity layer, but also resolve along strike changes in it, is left largely unsubstantiated.

A: We had already carried out this test. We have now included the results in two new figures. The slab anomaly recovery tests show that we should be able to resolve the transition depth, as long as it is shallower than ~100km.

In the checker board tests that are shown please clarify whether the checkerboards are limited to the proposed low velocity crust (e.g. 7 km in thickness), or are projected throughout the model area?

A: The checkerboard introduced only covers the top 15 km of the slab. We have now included vertical sections through the input and recovered anomalies to show their vertical extent (in supplementary Figures 9 and 10).

3. Local V_p minimum

This is a very specific and fine scale low velocity structure to interpret from a travel time tomography. A greater set of checkerboard tests (as described above) may go some way to assuring the reader that these features are resolved. But if you really want to test it, you could again test the model with a block model. Otherwise much of your discussion is resting on what untested and potentially unstable observations.

A: The checkerboard tests shown in supplementary figure 7 were carried out to show that we can resolve velocity variations in the crust of the slab. To strengthen our interpretation we have added one additional test with anomalies parallel to the trench.

Minor revisions:

Line 34: You mention that outer rise faults have a potentially significant role to play, but do not pick up on this point again. Do these structures have any role in the Lesser Antilles, or is there any evidence from your study?

A: This question was also posed by reviewer 1 above. Bend faults exist in the area as MCS profiles show evidence of displacement of the oceanic basement beneath the accretionary prism (Laigle et al., 2013). Since bend faulting happens landward of the deformation front (beneath the prism), these faults do not break the seabed, which may limit water penetration. Further north, where sediment cover is thin and the prism is much reduced in width and thickness, bend faults break the seabed seaward of the deformation front and are observed in multi-beam bathymetry data and in multi-channel seismic data (data presented at AGU Fall meeting 2016 show dipping reversed polarity reflectors penetrating beyond the slab Moho). We have now added some more discussion of this in the text.

Line 53: “a cold subduction” this is poorly worded. More precisely it is a subduction zone with a low thermal parameter.

A: We have reworded the sentence; however, we prefer to use simpler language in the introduction. The cold vs. hot subduction paradigm is well established therefore experts will know what we are talking about.

Line 59-61: It is unclear what you mean here. You mention two large thrust events. Are you suggesting that these compose of one sequence? This seems unlikely. Better to simply refer to the two historical events.

A: The two events are sometimes together referred to as a sequence. We have now rephrased the sentence.

Line 70-71: These statements need more detail or appropriate referencing. Which two large events? What evidence is there that the long term volcanic activity is more vigorous?

A: Estimates of volcanic production rates in the last 100,000 years indicate that the central part of the arc has erupted greater volumes of magma (Wadge, 1984). This is expressed in the size and elevation of the islands which are larger in this region (Guadeloupe, Dominica, Martinique). We have now added a reference.

Line 75: “one of the largest combined traveltimes databases.” This statement needs conditioning. Is this one of the largest travel time data sets in any geophysical study, or in a subduction zone setting, or just in the Lesser Antilles arc? Currently this is unclear.

A: In a subduction setting. The amount of earthquake data is not exceptional, what makes it unique is the combination of active source and local earthquakes and the dense 3D coverage. We have now added some context.

Line 84-85: This is unclear to me. What is a hanging 1D model? Do you mean that initial 1D model with a variable upper crust? Please clarify here and in the supporting info, and define the term 'hanging 1D model' with the non-seismologist reader in mind.

A: A 3D model built by “hanging” a single vertical V_P profile from a given interface. In our case we use sediment velocities beneath the seabed and up to the acoustic basement. Then we use a vertical velocity profile for the crust beneath this surface and up to the Moho depth. Finally, we add mantle velocities using a constant gradient. We have now expanded and clarified this in the methods section.

Line 107: I am confused why you are citing the Abers (2005) guided wave paper here, since you are not referring directly to this work. It would be better to more clearly state that a crustal low velocity layer is a phenomena noted in other subduction zones through guided wave studies as you do below.

A: We thought the Abers (2005) paper was a good reference for the general observation of a low- V_p layer, even though it focuses specifically on guided waves. We have removed it from here.

Line 109: “Our inversion provides one of the clearest images of a subducting slab crust”. Perhaps best making sure you say ‘tomographic images’ here as there are many far clearer receiver function images available in other arcs. Also consider broadening you referencing of other high resolution tomographic studies of the low velocities in subduction zones, particularly Zhang et al (2004, Geology) in Japan.

A: It’s true that migrated RF can provide sharper images of the slab crust. We have now edited the text following your suggestion.

Line 110-111: Please make it clear to the reader that all of the studies you refer to were conducted in other arcs. Your choice of papers imaging subducted low velocity crustal material in seems a little random as well.

Waveguide behavior – I am confused why you do not refer to Abers (2005) here, as you have already mentioned it earlier in the text. You should probably refer to Abers (2000) as this is the classic guided wave paper. There are also other more recent guided wave papers that consider subducting crust of similar age (e.g. Garth & Rietbrock 2014, GJI).

There are many many receiver function paper showing low velocity structure. So perhaps just refer to one or two significant papers. Yuan et al (200) is an early example, and Abers (2013) gives an overview of many arcs for example.

A: Thanks for pointing out some other important references. We were aware of them but had to cut many out because of limitations in the number of citations allowed. Our choice was perhaps a bit naïve. We have now added references to some of the works you suggested.

Line 134-136: This statement isn’t supported by the paper – can you refer to a study that suggests this or justify this statement otherwise?

A: We have now changed this statement and added a reference.

Line 138: I would strongly suggest referring to this as the Wadati-Benioff zone throughout the paper, as this is the most widely accepted and correct terminology.

A: Now changed.

Line 142: appears distributed > appears to be distributed.

A: Changed

Line 149-150: I must reiterate my strong concerns that this increase in velocity may be simply because you cannot resolve any low velocity feature that may be there.

A: See comments above on the new resolution tests. Also note that we see not just a fading of the low-Vp layer, but a transition to a high Vp anomaly. The velocities in the part of the slab crust beneath 100 km depth do not stay close to the starting model (as would be expected if resolution was low in a damped least squares inversion), but are higher. Note that there is no slab in the starting model at all.

Line 150-151: This is a very general statement. Given the thermal model you refer to later in the text, can you be more specific? Are the pressure and temperature conditions expected to be in the eclogite stability field?

A: This was intended as an initial general statement on which we expand later. The whole paragraph has been reworked.

Line 214-217: This is very similar to the high pore fluid pressure at the nose of the Japanese trench which of course supported slip during the Tohoku earthquake. Perhaps change the focus of this statement.

A: We have rewritten this sentence.

Figure 3: perhaps also show the mantle wedge seismicity here, as you link it with the proposed low velocity area in the text.

A: We show the mantle wedge seismicity in figure 4 (now figure 6). We prefer not to add it to figure 3 (now fig 5) as it is already a cluttered figure.

Figure 4: The Vp reduction is shown for the North of the area, when I believe it is seen in the south of the area (figure 3).

A: We separated the area north of the Marathon fracture zone from the area south of it. This was unclear, now clarified in the figure caption.

Reviewers' Comments:

Reviewer #2:

Remarks to the Author:

Paulatto and co-authors present a 3D model of P-wave velocity and V_p/V_s from the Antilles subduction zone that was created by the joint inversion of a remarkable combination of active and passive seismic data collected over this region over the last ~10-15 years. The resulting velocity models contain many very interesting features that greatly improve our understanding of processes at depth in subduction zones and will certainly be interesting to the readers of Nature Communications. The authors interpret the combined velocity structure and distribution of seismicity to make inferences about dehydration and seismic behavior. I reviewed an earlier version of this manuscript, and I find the revised version much improved in many respects. I recommend that it be published following minor revisions.

The authors have improved the discussion in the manuscript, clarified what features are resolved by their model and what inferences come from other co-located previous studies, and added a host of new resolution tests to justify their ability to recover different features described in the text (particularly the crust of the subducting plate). The author's rebuttal letter also clarified many of my questions/comments. My primary criticism of the previous version of this manuscript was that it was hard to say much about the megathrust interface and smaller scale features like evolution of subducted sediment with this model given the resolution (15x15x3 km). In the revised version, I think the authors have made it more clear what is known from other more spatially limited but higher resolution datasets in this area that provide a basis for interpreting features in their regionally extensive but lower resolution dataset.

I have a few more questions/comments, which I describe below, but I consider these minor.

Lines 55-56 – Consider giving range of sediment thicknesses (e.g., in the north and in the south) for this statement?

Line 56 – Change “vicinity” to “proximity”

Line 69 – Change “350 km long” to “350-km-long”

Lines 92-94 – What choices were made in Velest modeling in terms of relative weighting of velocity changes, station terms and earthquake locations?

Line 97 – change “2 km” to “2-km”

Line 107 – Change “a-priori” to “a priori”

Line 165 – The observation of distributed seismicity throughout the mantle wedge is very interesting and important, and the authors attribute this to mantle heterogeneity. In particular they mention stress changes in pyroxenite veins due to serpentinization of neighboring peridotites. Has such a process been documented elsewhere or experimentally? From Figure 4 (rows 3 and 4) it looks like the V_p in the mantle wedge is really quite high, which would preclude significant serpentinization. It was also my understanding that previous studies of mantle wedge seismicity here (Laigle et al) and in other subduction zones (e.g., Decca et al, in Sumatra) suggest that it is actually evidence that there is not much serpentinization. I think more discussion of the important observation of mantle wedge seismicity should be added to clarify the interpretation.

Line 177-179. As displayed, the figures do not convincingly show dipping planes of seismicity penetrating the slab, as described in these lines of the text. These need to be better demonstrated in the figures (e.g., with an inset with a zoom?) or removed. The yellow circle in row 4 of Figure 4 appears to just circle part of a broader cloud of seismicity.

Lines 291-299. I suggest omitting or greatly shortening this paragraph. As the authors write, there is large uncertainty in interpreting the Vp variations on the slab in terms of asperities, so they really cannot say much about it one way or the other. I also still think that the resolution of these models is not capable of resolving the scale lengths of features along the plate boundary that would be useful for such as discussion (particularly the vertical resolution of the model).

Right now, the manuscript ends with a whimper since the last point discussed is one with great uncertainty that is not particularly well addressed with the authors' new model, in my opinion. I would suggest instead to end the paper a few sentences on the global consequences of the very important new information this model gives about variations in hydration/dehydration downdip along and across the Antilles subduction zone.

Figure 4 – It is not clear to me that the earthquakes circled in yellow should be interpreted as dipping zones of earthquakes penetrating the slab, as mentioned above. Consider showing an inset with a zoom to demonstrate this.

Line 485 – Change “beach-balls” to “beach balls”. Or, better, consider calling them focal mechanisms instead of the more informal name of beach balls.

Reviewer #3:

Remarks to the Author:

Dear Editor,

I am happy to have re-reviewed this much improved revised manuscript. The authors have included a far wider range of resolution tests and checks, that make the results far more convincing to an informed reader with a seismological background. The discussion is also more focused on features that are well resolved, and more reasonably justified by the observations.

The manuscript still suffers from occasional odd phrasing, a few examples of which are outlined below. However, overall this improved manuscript makes some interesting and innovative observations, and is appropriate in scope and originality for nature communications.

Minor notes:

line 158 and 160: repetition of the 'sharp arcward termination' line 158 and 160 seems a bit repetitive – consider rewording

line 158 and 160: arcward > arc-ward .

line 185: is likely due to dehydration of the > is likely to be due to dehydration of the

References: Many of the references appear to be missing the year. Maybe check your referencing style.

REVIEWERS' COMMENTS:

Reviewer #2 (Remarks to the Author):

Paulatto and co-authors present a 3D model of P-wave velocity and Vp/Vs from the Antilles subduction zone that was created by the joint inversion of a remarkable combination of active and passive seismic data collected over this region over the last ~10-15 years. The resulting velocity models contain many very interesting features that greatly improve our understanding of processes at depth in subduction zones and will certainly be interested to the readers of Nature Communications. The authors interpret the combined velocity structure and distribution of seismicity to make inferences about dehydration and seismic behavior. I reviewed an earlier version of this manuscript, and I find the revised version much improved in many respects. I recommend that it be published following minor revisions.

The authors have improved the discussion in the manuscript, clarified what features are resolved by their model and what inferences come from other co-located previous studies, and added a host of new resolution tests to justify their ability to recover different features described in the text (particularly the crust of the subducting plate). The author's rebuttal letter also clarified many of my questions/comments. My primary criticism of the previous version of this manuscript was that it was hard to say much about the megathrust interface and smaller scale features like evolution of subducted sediment with this model given the resolution (15x15x3 km). In the revised version, I think the authors have made it more clear what is known from other more spatially limited but higher resolution datasets in this area that provide a basis for interpreting features in their regionally extensive but lower resolution dataset.

I have a few more questions/comments, which I describe below, but I consider these minor.

Lines 55-56 – Consider giving range of sediment thicknesses (e.g., in the north and in the south) for this statement?

Added

Line 56 – Change “vicinity” to “proximity”

Changed

Line 69 – Change “350 km long” to “350-km-long”

Changed

Lines 92-94 – What choices were made in Velest modeling in terms of relative weighting of velocity changes, station terms and earthquake locations?

We used the standard value of 0.5 for the weight of the S-waves relative to the P-waves. As far as I know station correction and source parameters cannot be weighted individually in Velest (without modifying the source code). The Velest inversion was used to get consistent initial locations and initial RMS residuals, to be used for declustering and decimation.

Line 97 – change “2 km” to “2-km”

Changed

Line 107 – Change “a-priori” to “a priori”

Changed

Line 165 – The observation of distributed seismicity throughout the mantle wedge is very interesting and

important, and the authors attribute this to mantle heterogeneity. In particular they mention stress changes in pyroxenite veins due to serpentinization of neighboring peridotites. Has such a process been documented elsewhere or experimentally?

There is general evidence that the forearc mantle wedge may be heterogeneous from analyses of exhumed ultramafic terranes (e.g. Santos et al., 2002, *Petrology*, v 43, pp 17-43). We are not aware of any experimental results. In the mantle wedge above the slab earthquakes may also be caused by hydrofracturing caused by fluid overpressure as suggested by Padron-Navarta et al (2010, *EPSL*), based on microstructural observations. We have added some additional discussion and references. Note to editor: these will take us above the recommended limit of 70 references, we hope this is not a problem.

From Figure 4 (rows 3 and 4) it looks like the V_p in the mantle wedge is really quite high, which would preclude significant serpentinization. It was also my understanding that previous studies of mantle wedge seismicity here (Laigle et al) and in other subduction zones (e.g., Decca et al, in Sumatra) suggest that it is actually evidence that there is not much serpentinization. I think more discussion of the important observation of mantle wedge seismicity should be added to clarify the interpretation.

V_p in the shallow mantle wedge is really quite variable, compare for example section 2 and section 4 from figure 4. We think serpentinization must be present, but only locally. We avoided focusing our interpretation too heavily on absolute V_p and V_p/V_s in the mantle wedge since these may be affected by anisotropy. We hope to be able to constrain seismic anisotropy in this region with new data that is currently being collected.

Line 177-179. As displayed, the figures do not convincingly show dipping planes of seismicity penetrating the slab, as described in these lines of the text. These needs to be better demonstrated in the figures (e.g., with an inset with a zoom?) or removed. The yellow circle in row 4 of Figure 4 appears to just circle part of a broader cloud of seismicity.

The southern cluster (row 4) is at an angle with the section so it doesn't look sharp. Perhaps this can be better seen in the supplementary movie, which provides a rotating 3D view of the earthquake distribution. We have added a reference to the supplementary movie here and in the caption of figure 4. We plan in the future to do double-difference relocation of the earthquakes which will further sharpen the seismicity.

Lines 291-299. I suggest omitting or greatly shortening this paragraph. As the authors write, there is large uncertainty in interpreting the V_p variations on the slab in terms of asperities, so they really cannot say much about it one way or the other. I also still think that the resolution of these models is not capable of resolving the scale lengths of features along the plate boundary that would be useful for such as discussion (particularly the vertical resolution of the model).

We would like to keep at least part of this discussion, in part because understanding megathrust properties was one of our initial main goals for this project, and in part because we think we have made some new observations that, although somewhat speculative, can contribute to the discussion. We have in part followed this suggestion by redacting and rearranging the final two paragraphs to make them a bit more uplifting.

Right now, the manuscript ends with a whimper since the last point discussed is one with great uncertainty that is not particularly well addressed with the authors' new model, in my opinion. I would suggest instead to end the paper a few sentences on the global consequences of the very important new information this model gives about variations in hydration/dehydration downdip along and across the Antilles subduction zone.

We have rearranged the last two sections, to end the paper with a summary observations and the most important implications.

Figure 4 – It is not clear to me that the earthquakes circled in yellow should be interpreted as dipping zones of earthquakes penetrating the slab, as mentioned above. Consider showing an inset with a zoom to demonstrate this.

See above.

Line 485 – Change “beach-balls” to “beach balls”. Or, better, consider calling them focal mechanisms instead of the more informal name of beach balls.

Changed

Reviewer #3 (Remarks to the Author):

I am happy to have re-reviewed this much improved revised manuscript. The authors have included a far wider range of resolution tests and checks, that make the results far more convincing to an informed reader with a seismological background. The discussion is also more focused on features that are well resolved, and more reasonably justified by the observations.

The manuscript still suffers from occasional odd phrasing, a few examples of which are outlined below. However, overall this improved manuscript makes some interesting and innovative observations, and is appropriate in scope and originality for nature communications.

Minor notes:

line 158 and 160: repetition of the ‘sharp arcward termination’ line 158 and 160 seems a bit repetitive – consider rewording

Changed to “sharp termination of seismicity towards the arc”

line 158 and 160: arcward > arc-ward .

Changed here and elsewhere in the text

line 185: is likely due to dehydration of the > is likely to be due to dehydration of the

Changed

References: Many of the references appear to be missing the year. Maybe check your referencing style.

Now fixed and reformatted according to Nature Communications guidelines